# Distinct IL-1α-responsive enhancers promote acute and coordinated changes in chromatin topology in a hierarchical manner

Sinah-Sophia Weiterer[1,†], Johanna Meier-Soelch[1,†], Theodore Georgomanolis[2,†], Athanasia Mizi[2,3], Anna Beyerlein[1], Hendrik Weiser[1], Lilija Brant[3], Christin Mayr-Buro[1], Liane Jurida[1], Knut Beuerlein[1], Helmut Müller[1], Axel Weber[1], Ulas Tenekeci[1], Oliver Dittrich-Breiholz[4], Marek Bartkuhn[5], Andrea Nist[6], Thorsten Stiewe[6,7] ⓘ, Wilfred FJ van IJcken[8] ⓘ, Tabea Riedlinger[9] ⓘ, M Lienhard Schmitz[7,9], Argyris Papantonis[2,3,*] ⓘ & Michael Kracht[1,7,**] ⓘ

## Abstract

How cytokine-driven changes in chromatin topology are converted into gene regulatory circuits during inflammation still remains unclear. Here, we show that interleukin (IL)-1α induces acute and widespread changes in chromatin accessibility via the TAK1 kinase and NF-κB at regions that are highly enriched for inflammatory disease-relevant SNPs. Two enhancers in the extended chemokine locus on human chromosome 4 regulate the IL-1α-inducible *IL8* and *CXCL1-3* genes. Both enhancers engage in dynamic spatial interactions with gene promoters in an IL-1α/TAK1-inducible manner. Microdeletions of p65-binding sites in either of the two enhancers impair NF-κB recruitment, suppress activation and biallelic transcription of the *IL8/CXCL2* genes, and reshuffle higher-order chromatin interactions as judged by i4C interactome profiles. Notably, these findings support a dominant role of the *IL8* "master" enhancer in the regulation of sustained IL-1α signaling, as well as for IL-8 and IL-6 secretion. CRISPR-guided transactivation of the *IL8* locus or cross-TAD regulation by TNFα-responsive enhancers in a different model locus supports the existence of complex enhancer hierarchies in response to cytokine stimulation that prime and orchestrate proinflammatory chromatin responses downstream of NF-κB.

**Keywords** chromatin topology; IL-8; interleukin-1; NF-κB; tumor necrosis factor-α

**Subject Categories** Chromatin, Transcription & Genomics; Immunology
**The EMBO Journal** (2020) 39: e101533

## Introduction

Inflammation is an evolutionarily conserved reaction to all forms of tissue injury and a major cause of human disease (Wallach *et al*, 2014). The cytokines interleukin-1 (IL-1) and tumor necrosis factor-alpha (TNFα) are potent mediators of inflammation across human tissues (Rock *et al*, 2010). Upon binding to cognate cell-surface receptors, IL-1 and TNFα initiate a cascade of cytosolic signaling events to eventually exert control over specific transcription factors (TFs) in the nucleus (Gaestel *et al*, 2009). A central upstream regulator in this scenario is the TAK1 protein kinase that activates the IKK, JNK, and p38 signaling pathways (Sakurai, 2012). All three pathways converge on regulating the nuclear concentration of TFs such as NF-κB and AP-1, thereby mediating cytokine-driven transcription at multiple responsive loci (Weber *et al*, 2010; Oeckinghaus *et al*, 2011; Zhang *et al*, 2017). While these modes of action are well established, a major unresolved question concerns the contribution of the non-coding genome to the coordinated IL-1/TNFα-triggered response in the three-dimensional (3D) space of the cell nucleus—i.e., how the various enhancers along chromosomes exert precise regulatory

1 Rudolf Buchheim Institute of Pharmacology, Justus Liebig University Giessen, Giessen, Germany
2 Center for Molecular Medicine Cologne, University of Cologne, Cologne, Germany
3 Department of Pathology, University Medical Center Göttingen, Göttingen, Germany
4 Research Core Unit Genomics, Institute of Physiological Chemistry, Medical School Hannover, Hannover, Germany
5 Institute for Genetics, Justus Liebig University Giessen, Giessen, Germany
6 Genomics Core Facility and Institute of Molecular Oncology, Philipps University Marburg, Marburg, Germany
7 Member of the German Center for Lung Research (DZL), Giessen, Germany
8 Center for Biomics, Erasmus Medical Center, Rotterdam, The Netherlands
9 Institute of Biochemistry, Justus Liebig University Giessen, Giessen, Germany
   *Corresponding author. Tel: +49 551 65734; E-mail: argyris.papantonis@med.uni-goettingen.de
   **Corresponding author. Tel: +49 641 9947600; E-mail: michael.kracht@pharma.med.uni-giessen.de
   †These authors contributed equally to this work

effects of different magnitudes in 3D space and over time to their cognate promoters during the inflammatory response.

Genomics approaches of increasing throughput now allow us to probe thousands of putative *cis*-regulatory elements across mammalian chromosomes (Long *et al*, 2016), also in response to proinflammatory cues (Ghisletti *et al*, 2010; Ostuni & Natoli, 2013; Kolovos *et al*, 2016). Genome-wide profiles for histone modifications, TF binding, and chromatin accessibility provide cell type-specific catalogues of enhancers correlated with gene activation or repression and with cell identity (Wang *et al*, 2008; Thurman *et al*, 2012; Bowman & Poirier, 2015). However, assignment of the activity and quantification of the strength of enhancers remains challenging and requires perturbation strategies in their native chromatin context (Nizovtseva *et al*, 2017; Furlong & Levine, 2018). In addition, enhancers operate under the spatial constraints of interphase chromosomes, which are now understood to be complex and often dynamic 3D entities. Mammalian chromosomes harbor numerous topologically associating domains (TADs) that mostly act to insulate enhancer function (Gibcus & Dekker, 2013; Yu & Ren, 2017). This type of spatial organization directs long-range regulatory interactions, and 3D chromatin topology can be a critical factor in inflammation (Xu *et al*, 2017). Chromosome conformation capture (3C) technology now allows mapping of such spatial interactions (Dekker *et al*, 2013), although it is often-times not possible to infer (dynamic) enhancer functions from the mere presence of chromatin loops, chromatin modifications, or open chromatin (Goldstein & Hager, 2018). Thus, the exact roles of enhancers, especially those acting in an apparently concerted manner on the same loci, remain poorly understood and need to be studied on a case-by-case basis via loss- and gain-of-function approaches to dissect their roles in the disease-relevant regulatory networks mediating the inflammatory response (Snetkova & Skok, 2018; Vermunt *et al*, 2019).

We recently identified a large number of IL-1α/TAK1-regulated enhancers in human epithelial cells characterized by inducible H3K27ac and NF-κB demarcation (Jurida *et al*, 2015). Here, we ask how different, yet concertedly activated, enhancers acting on the same responsive genes exert their rapid and precise regulatory function. We combine ATAC-seq, i4C-seq, and single-molecule RNA FISH with CRISPR/Cas9 microdeletions of discrete NF-κB binding elements or with CRISPR-guided transactivation to address this question. In brief, we show that IL-1α stimulation induces widespread remodeling of chromatin accessibility, in which the role of NF-κB, hitherto considered secondary to that of priming factors (Smale & Natoli, 2014), is both necessary and sufficient, and even capable of ectopically decondensing heterochromatin. Analysis of the prototypical *CXCL* chemokine locus on human chromosome 4 revealed a hierarchical relationship between two cytokine-induced enhancers. Remarkably, one of the enhancers exerts dominant control over the whole locus via both pre-established and dynamic contacts to gene promoters and other enhancers. Ultimately, the *IL8* enhancer controls secretion of the abundant IL-8 and IL-6 factors, while also supporting sustained IL-1α signaling to NF-κB and JNK/p38 MAP kinases. This suggests that enhancer interplay can be more complex than currently appreciated, involving a new type of "proinflammatory master enhancers" to robustly produce rapid and quantitative differences in gene expression.

## Results

### IL-1α stimulation drives widespread changes in chromatin accessibility via TAK1 and NF-κB

IL-1α stimulation of human KB epithelial carcinoma cells leads to an almost exclusive transcriptional induction of hundreds of genes initiating the proinflammatory cascade. Previously, we showed that induction is predominantly driven by NF-κB and that pharmacological inhibition of the TAK1 kinase suppresses most of the response (Jurida *et al*, 2015). To investigate dynamic changes of the chromatin landscape in response to IL-1α stimulation, we performed ATAC-seq (Buenrostro *et al*, 2013) in resting and IL-1α-stimulated KB cells in the presence or absence of the specific TAK1 inhibitor 5Z-7-oxozeaenol (TAKi). Widespread changes in accessibility along responsive loci such as *IL8* and *TNFAIP3* were observed (Fig 1A), but also genome-wide, with > 75,000 (76,687) ATAC-seq peaks emerging specifically in response to IL-1α stimulation. Importantly, accessibility at these IL-1α-induced peaks is abolished upon co-treatment with the TAK1 inhibitor and, thus, dependent on TAK1-mediated signaling (Fig 1B). Interestingly, > 50% (166,578) of all ATAC-seq peaks recorded in IL-1α-stimulated cells were also already accessible prior to cytokine induction, while ~15% (40,972) of these peaks remain largely accessible despite TAKi co-treatment (Fig 1B). Focusing on peaks that are rendered accessible in response to IL-1α, we found that ~9% overlap H3K27ac marks. Compared to untreated cells, these chromatin regions undergo remodeling to unmask NF-κB and AP-1 (FOS/JUN) binding motifs with significant enrichments (Fig 1C, *left*). Compared to TAKi- and IL-1α-co-treated cells, it was essentially only the NFKB1/2 and RELB motifs of the NF-κB family that showed diminished enrichment due to changes in local accessibility. This suggests that the TAK1 pathway controls not only nuclear translocation of TFs via inducible phosphorylation, but also chromatin remodeling at a specific subset of NF-κB binding sites (Fig 1C, *right*).

We then asked if these remodeled chromatin regions are related to the proinflammatory gene expression program. We found 2,051 genes in the vicinity of the H3K27ac-marked ATAC-seq peaks (within < 0.5 Mbp and in the same TAD), and these were highly associated with gene ontology terms relevant to proinflammatory responses (Fig 1D). Accordingly, accessibility at their TSSs was induced by IL-1α and reduced upon TAKi treatment (Fig 1E). We processed the ATAC-seq peaks assigned to these 2,051 genes via GARLIC, a computational tool designed to statistically link disease-relevant SNPs with putative *cis*-regulatory elements (Nikolic *et al*, 2017). This revealed significant association between SNPs in these accessible sites and multiple common inflammatory diseases (e.g., rheumatoid arthritis, psoriasis, systemic lupus erythematosus, and inflammatory bowel disease; Fig 1F). These results show that IL-1α-/TAK1-derived signals exercise broad and genome-wide control of disease-relevant non-coding elements.

Further independent evidence for a role of the TAK1-NF-κB pathway in chromatin regulation was obtained using a heterologous LacI-LacO reporter system (Jegou *et al*, 2009). The p65 subunit of NF-κB, a key downstream effector of TAK1, proved sufficient to open up chromatin locally in this assay (Appendix Fig S1A–C). The observed decondensation of an otherwise heterochromatic region was accompanied by concomitant reduction in H3K27me3 levels

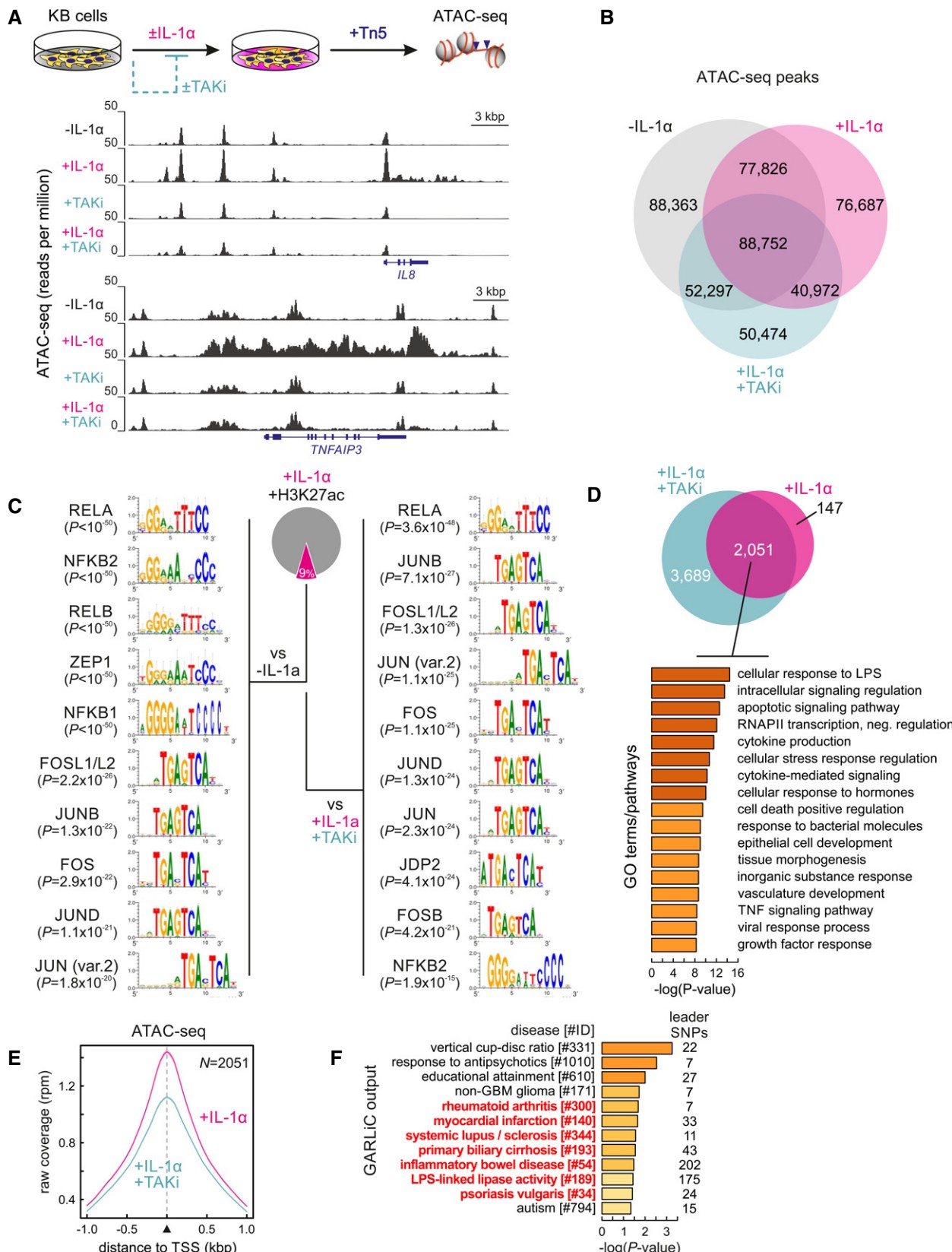

**Figure 1.**

◀

**Figure 1.   IL-1α-induced genome-wide changes in chromatin accessibility.**

A   KB cells were treated for 30 min with the TAK1 inhibitor 5Z-7-oxozeaenol (TAKi, 1 μM). Then, half of the cells were stimulated with IL-1α for 60 min resembling conditions previously described (Jurida *et al*, 2015). The cartoon illustrates the ATAC-seq experimental strategy (*top*). The genome browser views show representative changes in chromatin accessibility in two prototypical IL-1α-responsive loci (*IL8* and *TNFAIP3*).

B   Venn diagrams illustrating shared and condition-specific ATAC-seq peak regions in KB cells treated with IL-1α in the presence or absence of the TAK1 inhibitor (TAKi). Significant ATAC-seq peaks were determined using a more than twofold cutoff in read coverage over background together with a *q*-value of $< 10^{-4}$.

C   Analysis of TF motifs within ATAC-seq footprints in IL-1α-induced peaks overlapping H3K27ac (*Pie chart*) over those from uninduced (−IL-1α) or TAKi-treated cells (+IL-1α/+TAKi). Sequence logos and corrected discovery *P*-values for each motif are shown.

D   Gene ontology (GO) terms associated with the 2,051 genes in the vicinity of H3K27ac marks to which ATAC-seq peaks forming upon IL-1α induction and being sensitive to TAKi inhibition were assigned (*Pie chart*). Only genes within < 0.5 Mbp and the same TAD were included in this analysis.

E   Average profiles of ATAC-seq signals in the 2 kbp around the 2,051 TSSs from panel (D).

F   Diseases and traits associated with SNPs overlapping ATAC-seq footprints assigned to the 2,051 genes from panel (D); those with a known inflammatory component are highlighted (*red*).

and accumulation of active histone marks (H3K36ac) and phosphorylated isoforms of RNA polymerase II (Appendix Fig S1D). In addition, TNFα induced an increase in chromatin accessibility, as assessed at specific loci by formaldehyde-assisted isolation of regulatory elements (FAIRE), and this effect was suppressed by the knockout of *RELA* (Appendix Fig S1E). Taken together, these data define fundamental roles of factors (TAK1, p65) and *cis*-regulatory elements (NF-κB, AP-1) in controlling cytokine-driven changes in nucleosome density and chromatin accessibility in a concerted and rapid manner.

## IL-1α stimulation drives dynamic chromatin refolding in the *CXCL2* locus

We previously identified, by ChIP-seq in KB cells, four TAKi-sensitive enhancer regions flanking the prototypical chemokine locus of chromosome 4. They were characterized by IL-1-inducible H3K27ac and p65 recruitment (as shown in Fig EV1 and in Jurida *et al*, 2015), and we, therefore, used one of these enhancers downstream of the C*XCL2* locus, as a viewpoint to ask whether IL-1α-induced changes in chromatin accessibility also correlate with changes in spatial configuration. We obtained native spatial interactomes of the *CXCL2* promoter and enhancer by applying the "intrinsic (fixation-free) circularized chromosome conformation capture" (i4C) approach (Brant *et al*, 2016). This revealed involvement of the *CXCL2* promoter in a number of pre-established contacts with other IL-1α-inducible promoters and *cis*-regulatory elements throughout its locus. IL-1α stimulation for 1 h led to partial contact remodeling, mainly involving the responsive *CXCL3*, *CXCL1*, and *IL8* genes, as well as a number of enhancers and CTCF-bound sites. Most of these contacts were abolished upon TAKi treatment irrespective of IL-1α stimulation (Fig 2A, *top*), showing the relevance of basal and constitutive TAK1 activity in the process. The enhancer downstream of *CXCL2* was found looped to its cognate promoter already before IL-1α induction, which then allows for NF-κB binding to this promoter (Jurida *et al*, 2015) and also leads to TAK1-dependent contacts with the *CXCL1* and (less strongly) *IL8* gene promoters (Fig 2A, *bottom*). Meta-profiles of the average ATAC-seq and ChIP-seq signals at i4C contacts of either the *CXCL2* promoter or enhancer reveal that accessibility and H3K27ac and NF-κB/RNA polymerase II binding are generally increased by IL-1α stimulation and reduced by TAKi (Fig 2B). Taken together, our data indicate that IL-1α-induced chromatin remodeling renders NF-κB sites accessible, is sensitive to

TAK1 inhibition, and allows rapid spatial redistribution of contacts between IL-1α-responsive regulatory elements.

## Identification of hierarchically organized enhancers controlling the IL-1α response

To investigate the specific contribution of individual enhancers to gene expression in the extended *IL8/CXCL* locus, we decided to systematically delete those sites in the *IL8* and *CXCL2* proximal enhancers that we previously showed to most strongly bind the NF-κB p65 subunit in response to IL-1α treatment in KB and HeLa cells (Jurida *et al*, 2015) (Fig EV1). As HeLa (in our hands) are much more amenable to genetic perturbation, we used them to mutate individual NF-κB sites within the enhancers directly upstream of *IL8* or downstream of *CXCL2* by CRISPR/Cas9-mediated homozygous microdeletions of < 60 nt using pairs of sgRNAs (genomic positions indicated in Fig EV1). The resulting lines were validated by Sanger sequencing and are hereafter called Δp65$^{eIL8}$ or Δp65$^{eCXCL2}$ (Fig 3A).

However, before continuing with further experiments, we decided to revisit some key features of the IL-1α-responsive chemokine locus in both lines at the single-cell level. Both cell lines have been used in the IL-1 field for decades (Saklatvala *et al*, 1991; Bird *et al*, 1994; Freshney *et al*, 1994; Guesdon *et al*, 1997) and were originally isolated as separate epithelial carcinoma cell lines (Eagle, 1955a,b), but KB cells were later found to be a derivative of HeLa (Vaughan *et al*, 2017). While our HeLa and KB lines clearly differ morphologically (Appendix Fig S2A), they both strongly activate the chemokine cluster in response to IL-1α (as assessed by *IL8* RNA FISH) (Appendix Fig S2B and C). Compared to HeLa, KB cells show a more uniform IL-1α response at the single-cell level (Appendix Fig S2B and C). Moreover, DNA FISH reveals that KBs have two copies of chr. 4 on average, while HeLa cells mainly possess four copies (Appendix Fig S2D and E). Commercial short tandem repeat (STR) profiling from isolated DNA confirmed that the KB and HeLa cells used in this study are indeed identical in this aspect to original HeLa (Appendix Fig S2F) (Dirks & Drexler, 2013). We conclude that KB cells are a stable HeLa subclone that differs in copy number but otherwise shows a prototypical IL-1α-mediated activation of the *CXCL* chemokine locus.

In our enhancer-mutant HeLa lines, expression of all four chemokine mRNAs encoded by the *IL8/CXCL* locus, as well as of typical IL-1α-responsive genes on other chromosomes, was markedly decreased along a 180-min time course (Fig 3B). These data show that deletion of a single enhancer may affect not only the expression

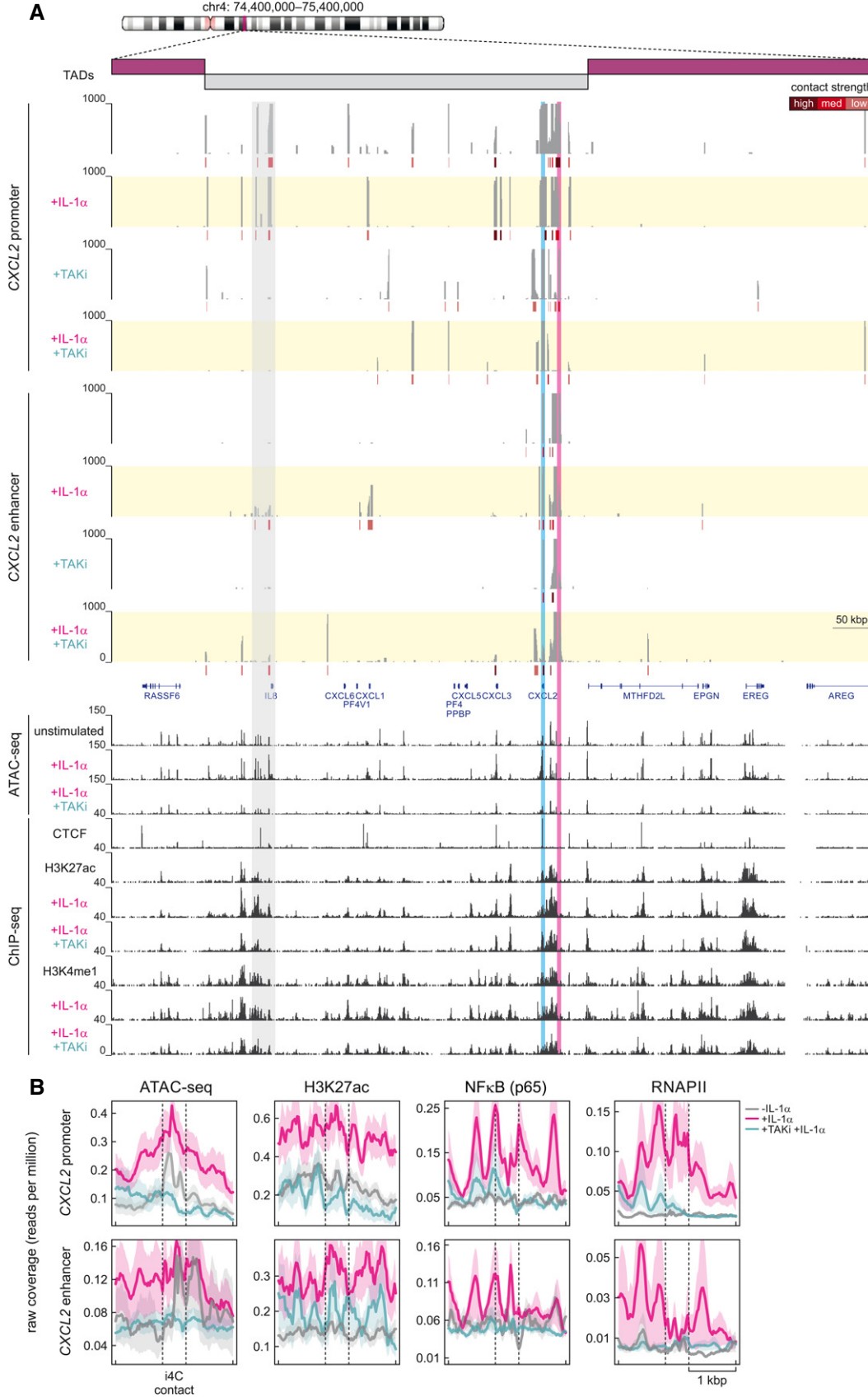

**Figure 2.**

**Figure 2. The IL-1α–TAK1 pathway regulates spatial chromatin interactions by the *CXCL2* locus.**

A   Cross-linking-free chromosome conformation capture (i4C) analysis was performed using chromatin from KB cells ± IL-1α stimulation for 60 min in the presence or absence of a TAK inhibitor (TAKi). Shown are i4C profiles in the 1 Mbp around the *CXCL2* locus on chromosome 4 (*ideogram*). Average read counts of two biological replicates are plotted, generated using the *CXCL2* promoter (*blue highlight*) or enhancer (*pink highlight*) as a viewpoint. The region of the *IL8* promoter/enhancer is also shown (*gray highlight*). Below each profile, significantly strong (*brown*), medium (*red*), or weaker interactions (*orange*) called via *foursig* software (Williams *et al*, 2014) are indicated. All profiles are shown aligned to gene models (*blue*) and CTCF ChIP-seq, as well as to H3K27ac and H3K4me1 ChIP-seq data from KB cells (GSE64224 + GSE52470) performed under the same conditions (Jurida *et al*, 2015). The breadth of topologically associating domains (TADs) in the locus is indicated above.

B   Meta-plots showing coverage of ATAC-seq (this study) and H3K27ac, p65, and RNA polymerase II (RNAPII) ChIP-seq signals (GSE64224 + GSE52470) at i4C fragments ± 1 kbp contacted by the *CXCL2* promoter or enhancer in KB cells ± IL-1α stimulation for 60 min in the presence or absence of a TAK inhibitor (TAKi).

Source data are available online for this figure.

of its cognate gene, but also the expression of all genes encoded in its locus. Interestingly, the effects of the $\Delta$p65$^{eCXCL2}$ deletion were consistently less dramatic than those of the $\Delta$p65$^{eIL8}$ one (e.g., for *CXCL1/3*; Fig 3B). These enhancer-mutant lines, as well as a line carrying both deletions ($\Delta$p65$^{eIL8 + eCXCL2}$), do not affect basal and IL-1α-inducible mRNA stabilities of *IL8* and *CXCL2* mRNAs, thereby ensuring that inhibition of gene activation manifests at the transcriptional level (Fig EV2A). Microarray experiments in these three p65-deletion lines revealed few changes at the whole-transcriptome level (compared to vector controls; Figs 3C and EV2B–D, Table EV1), and accordingly, there were also no changes in the 481 genes (out of 813 annotated genes) expressed from chr. 4 (Fig 3C). Together with the preserved integrity and copy number of chr. 4 (as assessed by DNA FISH in the $\Delta$p65$^{eIL8}$ cell line; Appendix Fig S2D, E), our data indicate that the microdeletions do not affect the overall structure of the chromosome. Nonetheless, we recorded a profound suppression of all major IL-1α-responsive genes (Figs 3C and EV2C). These are almost exclusively related to the proinflammatory response (Fig EV2D), and their suppression suggests a widespread effect of these two single-enhancer microdeletions on the deployment of the IL-1 transcriptional cascade.

To assess the impact of these enhancer microdeletions on chromatin modifications and NF-κB binding, we performed ChIP-qPCR for histone marks, NF-κB (p65), and RNA polymerase II at the promoters and enhancers of different IL-1α-responsive genes along a 180-min time course. Typically, p65 binding at the *IL8* and *CXCL2* promoters and enhancers will peak between 30 and 60 min post stimulation. This was almost abolished in $\Delta$p65$^{eIL8}$ cells, but in $\Delta$p65$^{eCXCL2}$, the *IL8* promoter and enhancer did still detectably bind p65 (Fig EV3A). Similarly, H3K27ac levels were strongly diminished only in $\Delta$p65$^{eIL8}$, while recruitment of initiating RNA polymerases (phosphorylated at Ser5 of their CTDs) was significantly reduced across the enhancer-mutant lines (Fig EV3A). Reduction of p65 and RNA polymerase loading, as well as of H3K27ac, was seen for other IL-1α-responsive genes in the same locus (*CXCL1* and *CXCL3*), but also for those on other chromosomes (*IL6*, *CCL20*, and *NFKBIA*). This reveals an unforeseen impact by a single enhancer on many inducible genes across the genome, in line with our microarray analysis. Again, this effect was more pronounced after deletion of the *IL8* rather than the *CXCL2* enhancer, suggesting a hierarchal relationship between these two regulatory *cis*-elements (Fig EV3B).

The aforementioned widespread effect should ultimately affect protein production—and in this case, the cells' secretome. We performed three types of analyses to assess the specificity and magnitude of changes in enhancer-mutant cells at the protein level,

along an extended time course after IL-1 stimulation. First, we confirmed the sustained suppression of *IL8* and *IL6* mRNAs in the $\Delta$p65$^{eIL8}$ mutant cells compared to cells depleted for p65 by CRISPR/Cas9 mutation of the *RELA* gene (Fig 3D, *upper graphs*). Specific ELISAs performed on the supernatants of the same cell cultures confirmed the suppression of secreted IL-8 and IL-6 proteins in the $\Delta$p65$^{eIL8}$ mutant to an extent comparable to the *RELA* knockout (Fig 3D, *lower graphs*). Second, profiling of 80 cytokines by semi-quantitative antibody arrays showed that IL-6 and IL-8 are indeed the most abundant IL-1α-induced secreted factors. This approach also identified CCL20 (MIP-3α) as another factor that is reduced in $\Delta$p65$^{eIL8}$ cells similarly to *RELA*-knockout levels (Appendix Fig S3; again in line with the RT–qPCR data in Fig 3B). Third, the fact that we observed no difference in the overall secreted proteome (assessed by silver staining of cell culture supernatants) or in the newly synthesized secreted proteome (assessed by *in vivo* puromycinylation of nascent peptide chains) between control cells and the *IL8* enhancer-mutant cells or p65-depleted cells (Fig 3E) argues that the suppression of these inflammatory regulators was strictly specific.

We next looked at the single-cell level and noted that nuclear translocation of NF-κB is less efficient in the presence of individual or combined enhancer deletions (Appendix Fig S4A, *top row*, and Appendix Fig S4B), with the accumulation of the NF-κB-driven *IL8* mRNA being strongly decreased 1 h post-stimulation and the *NFKBIA* mRNA moderately suppressed (Appendix Fig S4A, *middle/bottom row*s, and Appendix Fig S4C), again in line with our RT–qPCR data (Fig 3B). In addition, at the level of the NF-κB signaling cascade, its suppression in our enhancer-mutant lines is exemplified by reduced IκBα phosphorylation and degradation, as well as by reduced p65 phosphorylation in cell lysates (Appendix Fig S5A and B). Despite p65 protein and mRNA levels remaining unchanged (Appendix Fig S5A–C), more p65 was bound to IκBα protein in $\Delta$p65$^{eIL8}$ cells, thereby corroborating the inhibition of the cytosolic NF-κB signaling (Appendix Fig S5D). Moreover, activation of JNK and p38 MAPK was suppressed, revealing that these enhancers control all three major IL-1α-triggered pathways (Appendix Fig S5A and B), since the aforementioned IL-6, IL-8, and CCL20/MIP-3α (but also CXCL2/GRO-ß/MIP-2α via CXCR2) are direct and indirect regulators of canonical NF-κB and MAPK signaling (Heinrich *et al*, 2003; Manna & Ramesh, 2005; Ha *et al*, 2017; Jin *et al*, 2018). Finally, this enhancer-centric multilevel regulation is also supported by the finding that deletion of the NF-κB binding site within the *IL8* promoter ($\Delta$p65$^{pIL8}$) only affected *IL8* expression, but did not at all impact other IL-1α target genes or the activation of NF-κB signaling

(Fig EV4A–C). In contrast, *RELA*-knockout cells exhibit lower IκBα levels and essentially no IL-1α responsiveness, thus providing a control for the specificity of the other microdeletion phenotypes (Fig EV4B and D).

Taken together, these observations suggest a presumably indirect feedback mechanism that links the IL-1/IL-8 response to NF-κB nuclear relocalization (and MAPK activation), providing an explanation for the global effects of our enhancer mutants. Thus, HeLa lines

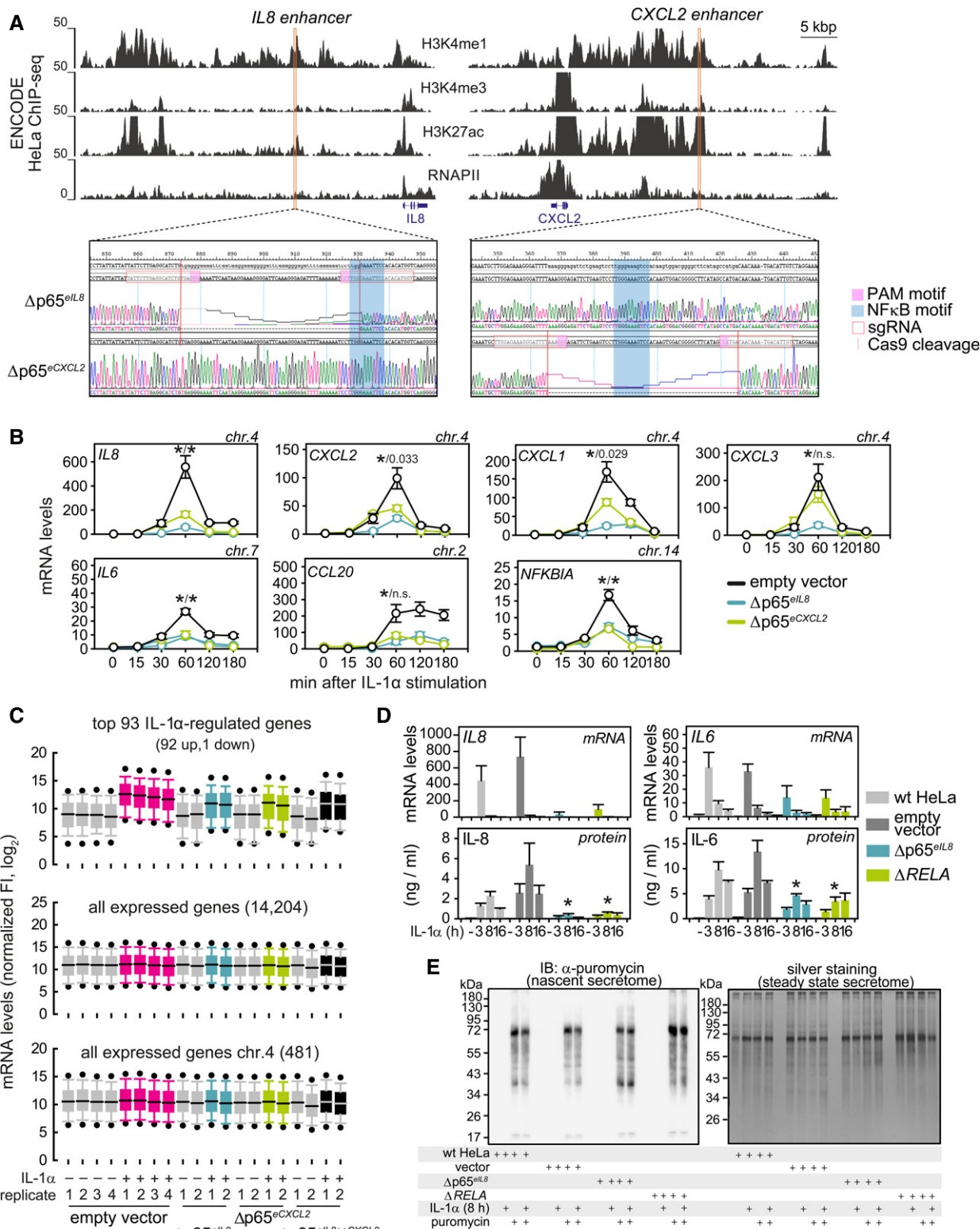

**Figure 3.**

◀

**Figure 3.  Deletion of NF-κB binding elements from the *IL8* and *CXCL2* proximal enhancers in HeLa suppresses inducible mRNA expression and secretion of IL-1α target genes.**

A   Genome browser views of the *CXCL2* and *IL8* chemokine loci on human chromosome 4 show H3K4me1, H3K4me3, H3K27ac, and RNA polymerase II ENCODE ChIP-seq profiles from HeLa-S3 cells relative to the *IL8* and *CXCL2* gene models (*blue*). The locations of the deleted NF-κB binding sites in their flanking enhancer regions are indicated (*orange*). Both loci were mutated using pairs of sgRNAs in stably transfected HeLa cell lines, and Sanger sequencing results of PCR-amplified genomic regions using DNA of both enhancer-mutant cell lines ($\Delta p65^{eIL8}$ and $\Delta p65^{eCXCL2}$) confirmed removal of 56 and 59 bp, respectively. Blue shades mark the targeted NF-κB binding sites.

B   mRNA levels of seven IL-1α-responsive genes in control (empty vector) or enhancer-mutant ($\Delta p65^{eIL8}$ and $\Delta p65^{eCXCL2}$) HeLa lines was assessed by RT–qPCR (mean levels ± SEM, normalized to *GUSB*; *n* = 4 (vector, $\Delta p65^{eIL8}$), *n* = 3 ($\Delta p65^{eCXCL2}$)) at the indicated times after IL-1α stimulation. *: significantly different to control; *P* < 0.01, unpaired, two-tailed Student's *t*-test.

C   Microarray gene expression analysis was performed in HeLa cells ± IL-1α stimulation for 60 min on control (empty vector; *n* = 4) and three p65 enhancer-deletion lines ($\Delta p65^{eIL8}$, $\Delta p65^{eCXCL2}$, and $\Delta p65^{eIL8+eCXCL2}$; *n* = 2). Differentially expressed genes were identified based on a moderated *t*-test (*P*-value < 0.05) and at least threefold change compared to the mean control levels (empty vector). The box plots show distribution of quantile-normalized mRNA expression values across all experimental conditions and cell lines. Gene sets (*from top to bottom*) represent IL-1α-regulated genes, all significantly expressed genes, and all mRNAs expressed from the genes of chromosome 4. Boundaries of the box indicate the 25th/75th percentiles, black lines within the box mark the medians, whiskers (error bars) indicate the 10th/90th percentiles, and black dots mark the 5th/95th percentiles. Additional analyses are provided in Fig EV2B–D. The complete data are provided in Table EV1.

D   Parental (wt), vector controls, *IL8* enhancer-mutant cells ($\Delta p65^{eIL8}$), or stable HeLa lines carrying CRISPR/Cas9-mediated mutations of the *RELA* gene ($\Delta RELA$) and therefore lacking p65 NF-κB (see also Fig EV4) were left untreated or stimulated with IL-1α as indicated. Then, total RNA from cell pellets and proteins from supernatants were analyzed by RT–qPCR and ELISA, respectively. *IL6* and *IL8* mRNA levels are depicted relative to the unstimulated vector controls (*upper panel*). IL-8 and IL-6 cytokine levels were normalized to total RNA, and concentrations are shown (*lower panels*). Data are from three independent experiments; shown are means ± SD.

E   Vector controls, *IL8* enhancer-mutant cells ($\Delta p65^{eIL8}$), or cells lacking p65 ($\Delta RELA$) were left untreated or were stimulated with IL-1α for 8 h in serum-free cell culture medium. After 7.5 h, half of the cells received puromycin for 30 min to label nascent polypeptides *in vivo* for monitoring ongoing translation (Iwasaki & Ingolia, 2017). Then, supernatants were harvested and proteins were precipitated and analyzed for newly synthesized polypeptides by immunoblotting using anti-puromycin antibodies (left *panel*) or for the entire stable secretome by silver staining (right *panel*). Shown is one out of two experiments yielding identical results.

carrying p65-binding site microdeletions reveal two hierarchically organized enhancers controlling gene expression in the early-responsive *IL8/CXCL* locus. While the *IL8* "master" enhancer displays a dominant effect on all genes in the locus, the *CXCL2* enhancer seems to be subordinate and to not generate as strong an effect. This suggests a unique hierarchy between distal enhancers controlling timing and amplitude of gene expression across an entire domain, but also in *trans*, in response to proinflammatory cues.

**Enhancer-deletion mutants reveal a hierarchy in spatial enhancer–promoter interactions**

The dominant effect that the $\Delta p65^{eIL8}$ deletion exerts on the regulation of all IL-1α-inducible genes in its locus could be explained by the spatial crosstalk among different promoters and enhancers. To assess this, i4C experiments were performed in wild-type and $\Delta p65^{eIL8}/\Delta p65^{eCXCL2}$ deletion cells using either the promoters or enhancers of *IL8* and *CXCL2* as viewpoints, which reside within the same TAD across cell types (Appendix Fig S6A). The *IL8* promoter is not found pre-looped to any other IL-1α-inducible promoter or enhancer within its TAD, but 1 h of IL-1α stimulation resulted in significant interactions with *CXCL2* and putative enhancers (Fig 4A, *top*). Using the *IL8* enhancer as a viewpoint allowed the detection of rapidly induced contacts between the *IL8* and *CXCL1* promoters (Fig 4A, *bottom*). In the $\Delta p65^{eIL8}$ line, interactions between the *IL8* promoter and enhancer and *CXCL2* are markedly diminished despite IL-1α stimulation, whereas $\Delta p65^{eCXCL2}$ cells still displayed rich interactomes with *CXCL1* and *CXCL2* (Fig 4A). Overall, i4C contacts by either the *IL8* promoter or enhancer are more enriched for H3K27ac than those in the $\Delta p65^{eIL8}/\Delta p65^{eCXCL2}$ cells (Fig 4B).

The *CXCL2* promoter showed few interactions with other genomic loci in unstimulated cells and developed strong contacts with the *CXCL1*, *CXCL3*, and *IL8* promoters after 1 h of IL-1α stimulation (Appendix Fig S7A, *top*). The *CXCL2* enhancer developed a

similar set of contacts to promoters post-stimulation, but also interacted with the *IL8* enhancer (Appendix Fig S7A, *bottom*). In $\Delta p65^{eCXCL2}$ cells, both the *CXCL2* promoter and enhancer interactomes are redirected away from IL-1α-inducible genes and regulatory elements (with the exception of the proximal *CXCL3* gene; Appendix Fig S7A). This is accentuated in i4C profiles of the *CXCL2* promoter in $\Delta p65^{eIL8}$ cells, although the promoter and enhancer of *CXCL2* remained associated in all replicates analyzed. Strikingly, the *CXCL2* and *IL8* enhancers studied here remain spatially associated upon IL-1α stimulation regardless of the genetic context of the cells tested (Appendix Fig S7A). Again, i4C contacts by either the *CXCL2* promoter or enhancer are on average more enriched for H3K27ac than those in $\Delta p65^{eIL8}/\Delta p65^{eCXCL2}$ cells (although less so than their *IL8* counterparts; Appendix Fig S7B). These data collectively reveal the importance of NF-κB-bound *cis*-regulatory elements in rewiring chromatin interactions. To also assess the contribution of the NF-κB p65 subunit in this cytokine-regulated process, we analyzed *RELA*-knockout cells, which do not show IL-1α-induced activation of *CXCL2/IL8* (as shown in Figs 3D and E, and EV4D and Appendix Fig S3C). Analysis of i4C interactomes from $\Delta RELA$ HeLa showed spatial associations between the *IL8* promoter and *CXCL2* promoter and enhancer, as well as with *CXCL3*, indicating that NF-κB is likely not a main driver of looping (Appendix Fig S8A).

In summary, all IL-1α-responsive promoters in the extended chemokine locus can be found interacting with one another in different combinations, but strong interactions between the *IL8* and *CXCL2* enhancers persist despite enhancer microdeletions or *RELA* knockout. This direct crosstalk, in conjunction with the differential i4C interactomes in the $\Delta p65^{eIL8}$ and $\Delta p65^{eCXCL2}$ lines, argues for a dominant role of the *IL8* "master" enhancer in the regulation of the whole locus. Such a hierarchical dominance explains the observed gene expression defects (Figs 3B–E and EV2C and D) on the basis of rapid changes (or lack thereof) in chromatin conformation.

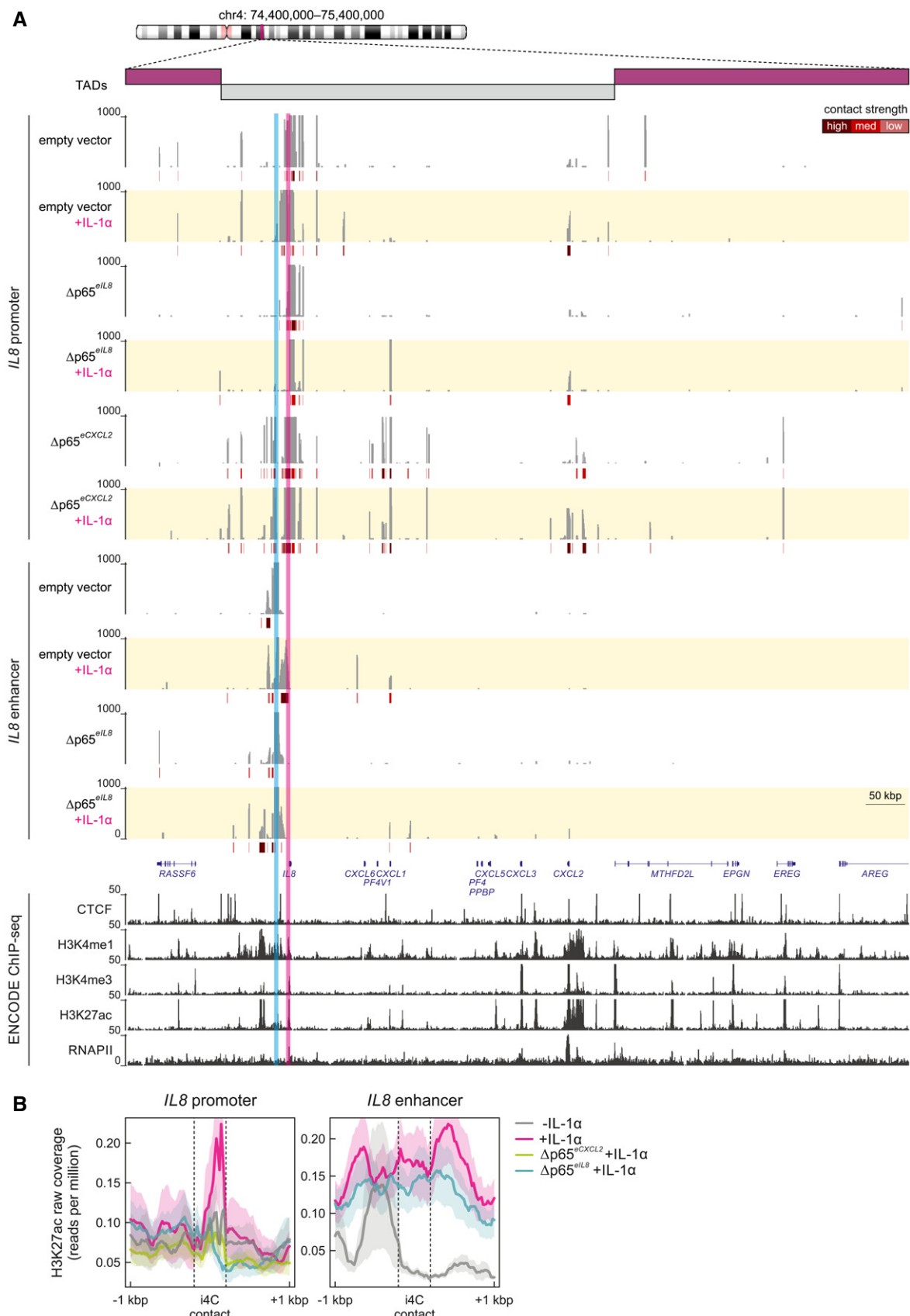

**Figure 4.**

**Figure 4.  Spatial chromatin interactions in the *IL8* locus are rewired by deleting p65-binding *cis*-elements within enhancers.**

A   i4C profiles in the 1 Mbp around the *IL8* locus on chromosome 4 (*ideogram*) from control (empty vector) and enhancer-mutant ($\Delta$p65$^{eIL8}$ and $\Delta$p65$^{eCXCL2}$) HeLa lines $\pm$ IL-1$\alpha$ stimulation for 60 min, generated using the *IL8* promoter (*pink highlight*) or enhancer (*blue highlight*) as a viewpoint. For the *IL8* promoter, the average of two biological replicates is shown, while for the *IL8* enhancer, data from one replicate are shown. Below each profile, significantly strong (*brown*), medium (*red*), or weaker interactions (*orange*) called via *foursig* are indicated. All profiles are shown aligned to gene models (*blue*) and to CTCF, H3K4me1, H3K4me3, H3K27ac, and RNA polymerase II ENCODE HeLa-S3 ChIP-seq profiles. The breadth of TADs in the locus is indicated above.

B   Meta-plots showing coverage of H3K27ac ChIP-seq signal at i4C fragments $\pm$ 1 kbp contacted by the *IL8* promoter or enhancer in control cells (empty vector) in the presence (*magenta*) or absence (*gray*) of IL-1$\alpha$ stimulation for 60 min, and in enhancer-mutant cells ($\Delta$p65$^{eIL8}$, *blue*; $\Delta$p65$^{eCXCL2}$, *green*) after IL-1$\alpha$ stimulation.

Source data are available online for this figure.

## Intronic RNA FISH reveals deficient *IL8* and *CXCL2* biallelic expression at the single-cell level

To provide orthogonal evidence for the mode of action suggested by our i4C results and to obtain a single-cell-level understanding of the enhancer-mutant effects, we performed RNA FISH with probes targeting the intronic (nascent) RNA produced by the *IL8* and *CXCL2* loci alongside of either *ACTB* or *IL8* mRNA (Fig 5A). Since transcriptional events occur in bursts, this approach allows quantification of transcriptional activity at individual transcription sites (Bartman *et al*, 2016). Quantification and statistical comparison of signals obtained in the presence/absence of IL-1$\alpha$ across all enhancer-mutant lines showed that microdeletion markedly reduces *IL8* and *CXCL2* transcription after 1 h of stimulation, without apparent hierarchy. Cells carrying either both enhancer deletions ($\Delta$p65$^{eIL8 + eCXCL2}$) or the full *RELA* knockout showed essentially no *IL8*/*CXCL2* activation (Fig 5B). We reasoned that this lack of a hierarchical effect was due to allelic discrepancies in *IL8* and *CXCL2* expression, as proinflammatory genes tend to be stochastically activated (Paixao *et al*, 2007; Apostolou & Thanos, 2008; Papantonis *et al*, 2010, 2012). Thus, we revisited our RNA FISH data stratifying for the fraction of cells showing colocalizing *IL8* and *CXCL2* intronic signals (i.e., transcribed from the same allele) reasoning that these events represent maximal IL-1$\alpha$-induced enhancer activation. Colocalization was significantly reduced in $\Delta$p65$^{eIL8}$ cells and essentially eliminated in $\Delta$p65$^{eIL8 + eCXCL2}$ cells, whereas the $\Delta$p65$^{eCXCL2}$ mutant had only a marginal effect on both biallelic expression and colocalization (Fig 5C). All effects were suppressed in $\Delta$*RELA* cells, in line with p65 driving inducible transcription across the chemokine locus (Fig 5C). Last, we performed intronic RNA FISH targeting *IL8* and *CXCL2* in the presence/absence of TAKi in HeLa, as well as in diploid retinal pigment epithelial (RPE-1) cells, verifying the transcriptional inhibition of mono- and biallelic expression of both loci (Fig 5D and E). This line of experiments supports the hierarchical relationship between the *IL8* and *CXCL2* enhancers also at the single-cell and nascent gene transcription levels.

## CRISPR-based activation of the *IL8* promoter and enhancer exerts discrete transcriptional effects

To validate our hierarchical model via an independent gain-of-function approach, we used the recently developed synergistic activation mediator (SAM) system (Konermann *et al*, 2015). This allows induction of single genes by specific targeting with an inactive Cas9 fused to the strong transactivation domains from NF-$\kappa$B (p65) and heat-shock factor 1 (HSF1) (Fig 6A). Targeting of this complex to

the *IL8* promoter resulted in its multi-fold activation, but induced no other gene in the entire locus (Fig 6B). However, targeting of the *IL8* enhancer using two different sgRNA pools activated not only *IL8*, but also *CXCL1* (while also mildly affecting *CXCL2/3*; Fig 6B). Repeating this approach with sgRNAs targeting the *CXCL2* enhancer and/or promoter failed to activate any other genes besides *CXCL2* (Fig 6B). These results are in full agreement with all previous data suggesting the functional dominance of the *IL8* "master" enhancer in controlling genes in the extended chemokine locus.

## A complex enhancer hierarchy in TNF$\alpha$-stimulated primary endothelial cells

To investigate whether a complex enhancer hierarchy also occurs in response to other cytokines, we tested two well-studied TNF$\alpha$-responsive loci on chromosome 14 (Papantonis *et al*, 2012; Kolovos *et al*, 2016) for changes in interactions in RELA-deficient HeLa cells. Using the *SAMD4A* promoter as viewpoint in i4C experiments, we confirmed previously published interactions in untreated and TNF$\alpha$-stimulated cells (Brant *et al*, 2016). The *BMP4* promoter interacted with the *SAMD4A* promoter in both unstimulated and TNF$\alpha$-treated cells (Appendix Fig S8B), despite the fact that they reside in two consecutive TADs (Appendix Fig S6B); again, interactions were largely p65-independent (Appendix Fig S8B). This prompted the question of whether enhancer hierarchies like those described above for the *IL8*/*CXCL2* locus also apply to cytokine-responsive loci in neighboring TADs.

To address this, we revisited i4C data from human primary endothelial cells (HUVECs) in the presence/absence of TNF$\alpha$ stimulation (Brant *et al*, 2016). Indeed, we could detect interactions between the *BMP4* and *SAMD4A* promoters irrespective of cytokine treatment (Fig EV5A, *top*). We previously showed that the enhancer upstream of *BMP4* (eBMP4) exerts mostly repressive effects to the gene, because it contains non-canonical NF-$\kappa$B binding sites and recruits the negative regulator JDP2 (Kolovos *et al*, 2016). On the other hand, the enhancer cluster in the first *SAMD4A* intron assists in gene activation (Larkin *et al*, 2012; Diermeier *et al*, 2014; Kolovos *et al*, 2016). We generated i4C data from eBMP4 and from the most TSS-proximal *SAMD4A* enhancer (eSAMD4A) after 60 min of TNF$\alpha$ stimulation and observed that each enhancer contacts its cognate gene promoter, but eSAMD4A also strongly contacts the *BMP4* promoter (Fig EV5A, *bottom*). We then reasoned that deletion of these enhancers would differentially affect the response of *BMP4* and *SAMD4A* to TNF$\alpha$, with the former being typically suppressed and the latter markedly induced (Kolovos *et al*, 2016). Using CRISPR/Cas9 deletions, we removed the whole eBMP4 or eSAMD4A regions in > 30% and ~12% of alleles in a heterogeneous HUVEC

population (as it is especially challenging to obtain single-cell-derived pools; Fig EV5B and C). Despite not being present in all alleles, these deletions caused effects on both genes: Deleting eBMP4 leads to the partial derepression of *BMP4*, while also suppressing

*SAMD4A* (Fig EV5B). Deleting eSAMD4A negatively affects the TNFα-mediated induction of *SAMD4A*, while also suppressing *BMP4* expression (the TNFα-inducible *CXCL2* gene provides a control; Fig EV5C). Notably, the enhancer region in the deleted eBMP4 allele

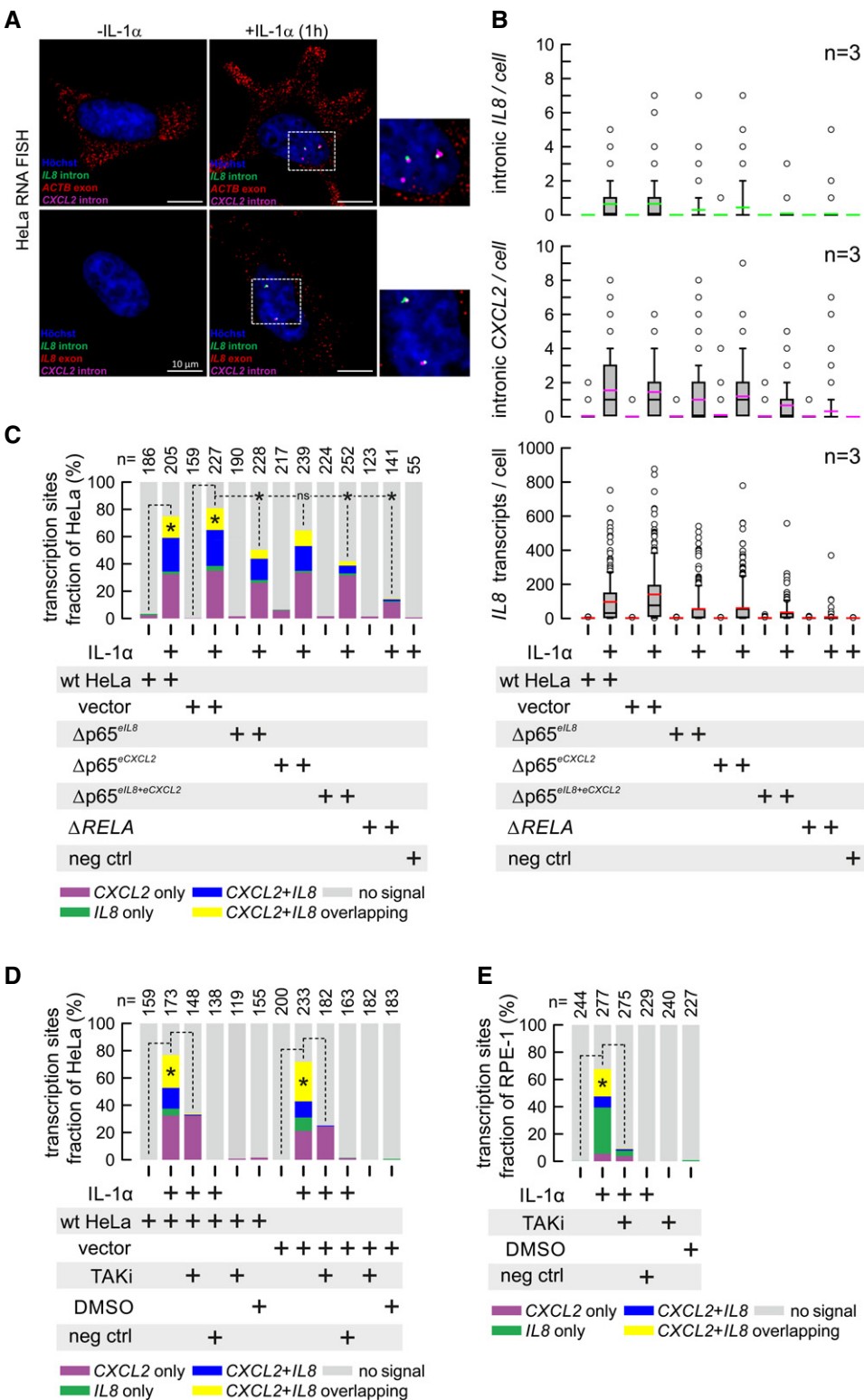

Figure 5.

Figure 5.  **Intronic RNA FISH reveals reduced concomitant biallelic and colocalizing chemokine expression in enhancer-mutant HeLa.**

A    Representative triple RNA FISH images from HeLa cells ± IL-1α stimulation for 60 min. Mature mRNAs (*IL8*, *β-actin*; *red*) and intronic RNAs (*CXCL2*, *purple*; *IL8*, *green*) are detected against nuclei stained with Hoechst 33342 (*blue*). Typical foci marking individually labeled *IL8/CXCL2* transcription sites or merged signals indicating co-transcription and spatial proximity are enlarged (*inset*). Scale bar: 10 μm.

B    Quantification of RNA FISH signals from parental (wt), control (vector), p65-deletion (Δp65^eIL8 and Δp65^eCXCL2), or p65-knockout (ΔRELA) HeLa lines ± IL-1α stimulation for 60 min. Negative controls (neg ctrl) indicate samples from IL-1α-stimulated control cells in which RNA FISH was performed using pre-amplifier, amplifier, and label probe mixes, but omitting the specific probe sets for *IL8* or *CXCL2*. These samples were used to define unspecific signals. Data from three independent experiments are pooled and plotted. The box plots show the distributions of FISH signals. Boundaries of the box indicate the 25th/75th percentiles, black lines mark medians, and colored lines mark means, respectively. Whiskers (error bars) indicate the 10th/90th percentiles, and circles mark all remaining outliers.

C    The data from panel (B) were used to separately quantify the fraction of cells with mono- or biallelic *IL8* or *CXCL2* intronic RNA expression (*purple, green, blue colors*), as well as the extent of colocalizing (overlapping) intronic RNA FISH signals in individual cells, indicating simultaneously activated transcription sites on the same allele (*yellow colors*). The total numbers of cells analyzed are shown above each bar. Data are depicted relative to the total number of analyzed cells. *$P < 0.05$; Fisher's exact test.

D    Parental (wt) or control HeLa cells (vector) were treated with the TAK1 inhibitor (TAKi) or solvent (DMSO) for 30 min ± IL-1α stimulation for 60 min. Intronic RNA FISH was performed in three independent experiments and quantified as in panel (C). The total numbers of cells analyzed are shown above each bar. *$P < 0.05$; Fisher's exact test.

E    As in panel (D), but for human pigmented retinal epithelial cells (RPE-1) treated with the TAK1 inhibitor (TAKi) or solvent only (DMSO) for 30 min ± IL-1α stimulation for 60 min.

Data information: In panels (D, E), data are pooled from three independent experiments.

still interacts with the *BMP4* promoter (Fig EV5A, *bottom*). Finally, to further support this functional crosstalk, we adapted an approach similar to the "multi-contact" 4C approach (MC-4C; Allahyar *et al*, 2018) on the basis of i4C and by coupling it to PacBio long-read sequencing (Fig EV5D and E). We generated MC-i4C interactomes for the *SAMD4A* promoter and enhancer, as well as for the *BMP4* enhancer. They appear to contribute to a higher-order chromatin hub, which would allow for the observed functional interference and co-regulation (Fig EV5F). Thus, we obtained evidence from diploid primary cells on the existence of complex enhancer hierarchies in response to cytokine stimulation, whereby two enhancers separated by > 0.5 Mbp confer unequal regulation across a TAD boundary in response to inflammatory stimuli.

## Discussion

Genetic and structural variation at enhancers is increasingly discussed as an underlying cause for disease, such that the term "enhanceropathies" has now been coined (Chen *et al*, 2018; Patten *et al*, 2018; Rickels & Shilatifard, 2018). While this concept evolved from cancer studies, emerging evidence supports a role of chromatin architecture and the non-coding genome also in inflammatory responses and the immune system (Smale & Natoli, 2014; Smale *et al*, 2014). In this context, enhancers have been shown to control differentiation and transcriptional responses in innate immune cells. For example, lymphocytes from lupus patients have altered histone quantitative trait loci (hQTLs) linking quantitative changes in enhancer PTMs to the disease (Pelikan *et al*, 2018). Likewise, enhancers of colon epithelial cells isolated from patients with inflammatory bowel disease are enriched in disease-linked SNPs (Boyd *et al*, 2018). Thus, understanding how distinct enhancers may synergistically or antagonistically control particular gene loci via detailed perturbations represents an eminent biomedical goal.

To this end, we present here new data on how a hierarchical regulatory relationship between two single cytokine-activated enhancers controls the prototypic chemokine locus expressing *CXCL1-3* and *IL8* (*CXCL8*) in human epithelial cells. The coordinated and quantitative expression of these genes is of high pathophysiological relevance, as the formation of chemokine gradients is an indispensable step for leukocyte recruitment to any inflamed tissue, thus constituting a fundamental process of innate immunity (Tan & Weninger, 2017). Chemokines are also key factors of the inflammatory tumor microenvironment, in which IL-8 is specifically known to also promote angiogenesis (Liu *et al*, 2016). First, we use ATAC-seq to show increased and coordinated chromatin accessibility around NF-κB binding sites, in line with previous nucleosome positioning data suggesting the immediate-early priming of the chromatin landscape for inflammatory stimulation (Diermeier *et al*, 2014). Accessible ATAC-seq footprints are also rich in AP-1 motifs (FOS, FOSL1/L2, c-JUN/JUND/JUNB), on top of the various NF-κB ones. We previously showed that these factors bind to *IL8/CXCL2* enhancers in an IL-1α/TAK1/p65-dependent manner and that *RELA* (p65) knockdown prevented AP-1 loading and gene activation. In contrast, knockdown of c-Fos or JunD only weakly affected *IL8* or *CXCL2* expression and had no effect on p65 enhancer binding (Jurida *et al*, 2015). Together, these data suggest that AP-1 coordinates with NF-κB for recruitment to chromatin and then plays a role in IL-1α-mediated chromatin folding rather than in transcription. Along these lines, such cooperativity has been shown for AP-1 contributing to static and dynamic chromatin looping in developing macrophages (Phanstiel *et al*, 2017), for the NF-κB-assisted loading of STAT3 at IL-1α-activated enhancers in hepatocytes (Goldstein *et al*, 2017; Vierbuchen *et al*, 2017; Madrigal & Alasoo, 2018), as well as for the role of AP-1 in chromatin accessibility and enhancer selection in murine fibroblasts or iPSCs (Goldstein *et al*, 2017; Vierbuchen *et al*, 2017; Madrigal & Alasoo, 2018).

Additional data from heterologous reporters and FAIRE now support a direct role of p65 NF-κB in changing the nucleosomal landscape at activated loci. This again may require cooperation with preloaded AP-1 factors, as so far there is little evidence to suggest that NF-κB alone acts as a pioneering factor (Diermeier *et al*, 2014; Monticelli & Natoli, 2017). Importantly, our work demonstrates how the TAK1 kinase may integrate all these processes, as its pharmacological inhibition suppresses factor recruitment, chromatin folding, and gene activation, most likely due to the relevance of TAK1 for activation of NF-κB and also MAPK signaling cascades that finally trigger activity of additional TFs such as AP-1 (Jurida *et al*, 2015).

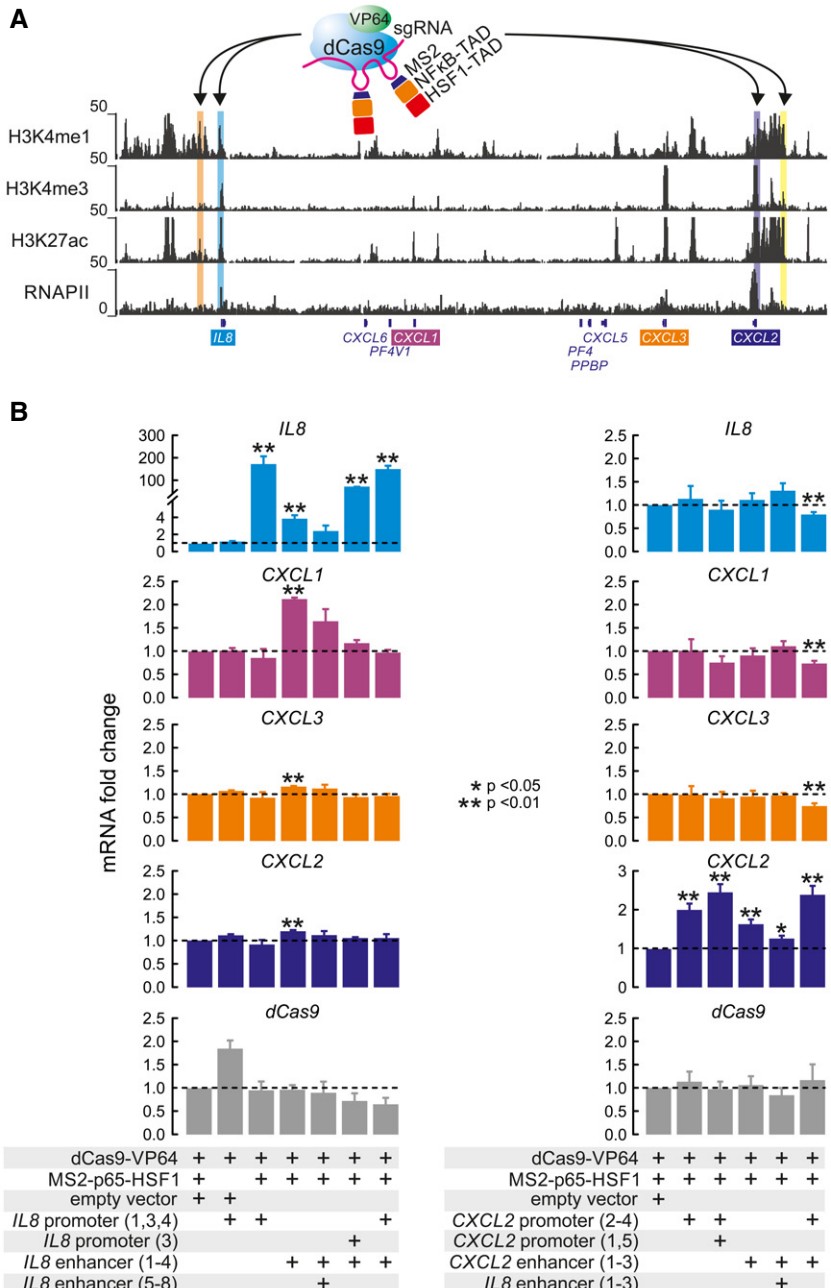

**Figure 6. Heterologous activation of the *IL8* enhancer triggers *IL8* and *CXCL1* expression.**

A  A CRISPR activation (CRISPRa) strategy was applied to test enhancer functions individually. This approach involves a "dead" Cas9 (*blue*) and VP64 (*green*) fusion protein that recruits the NF-κB (*orange*) and HSF1 (*red*) transactivation domains via MS2 recognition of two stem loops in the sgRNA scaffold (*magenta*). These complexes were targeted to the *IL8* or *CXCL2* enhancers and promoters (highlights) via different sgRNA pools in HeLa. The position of individual sgRNAs used for CRISPRa is shown in more detail in Fig EV1.

B  Left bar graphs: Wild-type HeLa cells were transiently transfected with different combinations of plasmids encoding the "dead" Cas9-VP64 fusion protein, the MS2-p65-HSF1 fusion protein, and empty sgRNA vector or versions containing sgRNAs targeting the *IL8* enhancer or promoter. Twenty-four hours post-transfection, cells were lysed and total RNA was analyzed for expression changes of the indicated genes compared to samples carrying dCas9-VP64 and MS2-p65-HSF1 fusions, but no sgRNAs. Right bar graphs: The same experiments were performed using sgRNAs targeting the *CXCL2* enhancer and promoter.

Data information: All data are from four independent transfections. Shown are mean values ± SEM. *P*-values are derived from unpaired *t*-tests comparing every condition against cells expressing all transactivators but lacking sgRNAs (first lane in each graph). Only significant differences are marked by asterisks.

Remarkably, the perturbation of chemokine enhancers affected the IL-1α response at multiple levels. In particular, the *IL8* enhancer seems to exert a dominant function in this inflammatory pathway. At this point, we can offer several explanations for this effect. First, data from this study suggest that the enhancer regulates rapid autocrine feedback loops involving the most abundant

secreted factors IL-8, IL-6, and CCL20/MIP-3α, which are known to support various inflammatory signaling pathways (Heinrich *et al*, 2003; Manna & Ramesh, 2005; Ha *et al*, 2017; Jin *et al*, 2018). It has been shown that IL-8 activates nuclear translocation of NF-κB in HeLa cells, which is consistent with the downregulation of NF-κB signaling observed in the *IL8* enhancer-mutant cells (Manna & Ramesh, 2005). Second, it is possible that our enhancer mutations indirectly affect the scaffolding function of a new class of long non-coding (lnc)RNAs that connect enhancers with promoters *in cis* through the WDR5-MLL1 protein complex within the *CXCL* chemokine locus (Fanucchi *et al*, 2019). In the TNFα response, these lncRNA–protein interactions apparently also activate transcription of multiple immune genes, adding to the idea of an upstream function of the chemokine locus in the inflammatory response. However, that mechanism is disparate to the one reported here, as it neither involves NF-κB components nor does it change chromatin looping in the *CXCL* locus (Fanucchi *et al*, 2019). Third, based on the highly coordinated activation of IL-1α target genes and their suppression upon enhancer perturbation, the chemokine locus may rapidly extrude and loop out of its chromosome territory to contact other loci (*IL6, CCL20, NFKBIA*) *in trans*. So far, we did not detect such contacts in our i4C data, but these interactions may occur less frequently, more stochastically, and in a burst-like fashion and, therefore, escaped detection. However, the strong phenotypes described here can be surveyed in future experiments to reveal the existence and functions of such inter-chromosomal contacts by more sensitive, emerging methods (Maass *et al*, 2019).

Once chromatin accessibility in response to IL-1α is ensured, multiple spatial contacts were seen to form natively in the *IL8/CXCL* locus, but these only weakly depend on chromatin binding by NF-κB (which also holds true for the *BMP4/SAMD4A* loci). Notably, we observe pre-established and persistent interaction between the *IL8* and *CXCL2* enhancers flanking the locus. Both these enhancers are rapidly activated in response to IL-1α, but their detailed analysis revealed that despite their identical activation patterns, the former has a dominant effect for gene activation, whereas the latter essentially only controls its nearby *CXCL2* gene promoter. This is strictly modulated by the "master" enhancer element, since neither heterologous activation nor CRISPR/Cas9 deletion of the *IL8* promoter affected any other gene in the locus.

Somewhat similar hierarchies were recently proposed for individual enhancers within the *MYO1D* and *SMYD3* "super-enhancer" clusters (Huang *et al*, 2018). Of course, these are multiple enhancers controlling a single target gene, but they could be subdivided into "hub" and "non-hub" enhancers on the basis of their CTCF/cohesin association and disease-relevant SNP content to show that hub enhancers are principal contributors to gene activation (Huang *et al*, 2018). Here, using primary endothelial cells and an i4C variant that allows identification of multi-way contacts, we can propose a spatial enhancer crosstalk taking place even across a TAD boundary and allowing formation of a "factory" that permits complex regulatory hierarchies to unfold.

In summary, the identification of hierarchically organized and spatially co-associated signal-responsive enhancers highlights the importance of chromatin-based mechanisms for inflammatory gene responses and adds a perhaps unforeseen layer to their regulation in cell nuclei.

# Materials and Methods

### Cell lines and cytokine treatments

HeLa cells (Handschick *et al*, 2014), KB cells (Jurida *et al*, 2015), and hTERT-immortalized retinal pigment epithelial cells (hTERT RPE-1, ATCC® CRL-4000™, a kind gift from Zuzana Storchova, Martinsried, Germany) were maintained in Dulbecco's modified Eagle's medium (DMEM) or DMEM high glucose (GlutaMAX supplemented with pyruvate) or DMEM/F12 (RPE-1), complemented with 10% fetal calf/bovine serum (FCS or FBS from PAN Biotech), 2 mM L-glutamine (HeLa and KB cells), 100 U/ml penicillin, and 100 μg/ml streptomycin. HeLa and KB cells were tested for mycoplasma with Venor®GeM Classic kit (Minerva Biolabs), and their identity was confirmed by commercial STR testing at the DSMZ-German Collection of Microorganisms and Cell Cultures (https://www.dsmz. de/dsmz). Stable pools of cell lines generated by transfections of the pX459-based CRISPR/Cas9 constructs were selected and maintained in puromycin (1 μg/ml). Prior to all experiments, puromycin was omitted for 24 h and IL-1α (10 ng/ml) was added directly to the cell culture medium. TAK1 inhibitor (5Z-7-oxozeaenol) was always added 30 min prior to further treatments. HUVECs from pooled donors (Lonza) were maintained in complete Endopan-2 (PAN Biotech) supplemented with 3% FCS and serum-starved in 0.5% FCS overnight before TNFα treatment (Peprotech; 10 ng/ml).

### Cytokines, inhibitors, and antisera

Human recombinant IL-1α was a kind gift from Jeremy Saklatvala (Oxford, UK) or was prepared in our laboratory as described and used at 10 ng/ml final concentration in all experiments (Rzeczkowski *et al*, 2011). Human recombinant TNFα (used at 10 ng/ml final) was from ImmunoTools. The following inhibitors were used: actinomycin D (Sigma-Aldrich, #A1410), leupeptin hemisulfate (Carl Roth, #CN33.2), microcystin (Enzo Life Sciences, #ALX-350-012-M001), pepstatin A (Applichem, #A2205), PMSF (Sigma-Aldrich, #P-7626), and 5Z-7-oxozeaenol (Tocris Bioscience, #253863-19-3, or, Enzo Life Sciences, #66018-38-0). The inhibitors actinomycin D (5 μg/ml final) and 5Z-7-oxozeaenol (1 μM final) were dissolved in DMSO prior to use and applied at dilutions > 1:1,000. Appropriate DMSO concentrations served as vehicle controls in experiments using small-molecule inhibitors. Pepstatin A, PMSF, and microcystin were dissolved in ethanol and leupeptin as well as the protease inhibitor cocktail tablet in dH$_2$O. Other reagents were from Sigma-Aldrich or Thermo Fisher Scientific, Santa Cruz Biotechnology, Jackson ImmunoResearch, or InvivoGen and were of analytical grade or better. Primary antibodies against the following proteins or peptides were used: anti-β-actin (Santa Cruz, #sc-4778), anti-CRISPR-Cas9 (Abcam, #191468), anti-CTCF (Millipore, #07-729), anti-FLAG (Sigma-Aldrich, #F1804), anti-H3 (Abcam, #ab1791), anti-H3K4me1 (Abcam, #ab8895), anti-H3K27ac (Diagenode, Pab-174-050), anti-H3K27me3 (Millipore, #07-449), anti-H3K36ac (Diagenode, #C15410307), anti-(PS32)-IκBα (Cell Signaling, #2859), anti-IκBα (Cell Signaling, #9242), anti-P(T183/Y185)-JNK (Cell Signaling, #9251), anti-JNK (Santa Cruz, #sc-571), anti-P(S536)-p65 (Cell Signaling, #3033), anti-p65 (Santa Cruz, #sc-372; #sc-8008), anti-P(T180/Y182)-p38 (Zymed, #36-8500), anti-

p38 (Cell Signaling, #9212), anti-puromycin (3RH11; Kerafast, #EQ0001), anti-P(S2)-RNA Pol II (Abcam, #ab5095), anti-P(S5)-Pol II (Abcam, #ab5131), anti-RNA-Pol II (Millipore, #17-620), anti-tubulin (Santa Cruz, #sc-8035), and normal rabbit IgG (Santa Cruz, #sc-2027; Cell Signaling #2729). Secondary antibodies used for immuno-FISH and immunoblotting were DyLight 488-coupled anti-mouse IgG (ImmunoReagent, #DkxMu-003D488NHSX), goat anti-rabbit IgG Cy3 (Diagenode, #111-165-003), HRP-coupled anti-mouse IgG (Dako, #P0447), HRP-coupled anti-rabbit IgG (Dako, #P0448; Thermo Fisher Scientific, #31460), and TrueBlot HRP-conjugated anti-rabbit IgG (Rockland, #18-8816-31).

## Plasmids and transient or stable transfections

The following plasmids were gifts or were obtained commercially: pSpCas9(BB)-2A-Puro (pX459; Addgene (#48139)), pSpCas9(BB)-2A-Puro V2.0 (pX459V2.0, Addgene (#62988)), lenti-sgRNA(MS2)-zeo backbone (Addgene, #61427), lenti dCAS9-VP64_Blast (Addgene, #61425), and lenti MS2-P65-HSF1_Hygro (Addgene, #61426). The following vectors were cloned: pX459sg1Δp65, pX459-sg1IL8Promoter, pX459-sg1/2/3IL8Enhancer, pX459-sg1/2CXCL2Enhancer, pX459V2.0-sg2IL8Promoter, lenti-sgRNA(MS2)-zeo-sg1/2/3/4IL8Promoter, lenti-sgRNA(MS2)-zeo-sg1/2/3/4/5/6/7/8IL8Enhancer, lenti-sgRNA(MS2)-zeo-sg1/2/3/4/5CXCL2Promoter, and lenti-sgRNA(MS2)-zeo-sg1/2/3CXCL2Enhancer. For mRNA expression analysis and immunoblotting experiments, HeLa cells were seeded at $0.5 \times 10^6$ cells per 60-mm dish or $1.5 \times 10^6$ cells per 100-mm dish. For single-cell analysis (RNA FISH), cells were seeded at 9,000 cells per slot in µ-slides VI (Ibidi). For stable sgRNA transfections, HeLa cells were transfected by the calcium-phosphate method and pools of cells were selected in complete medium with 1 µg/ml puromycin (Kracht Lab). For CRISPR/Cas9-mediated knockout of p65 in HeLa cells (Schmitz Lab; Appendix Fig S1), the non-transfected cells were eliminated by the addition of puromycin (1 µg/ml) 1 day after transfection for 48 h. After approximately 1 week, single-cell-derived clones were picked and further analyzed for expression of p65 and FLAG-Cas9.

## CRISPR/Cas9-mediated deletion of enhancer elements and validations

In HeLa cells, the classical CRISPR-Cas9-system was used to specifically delete elements within enhancers. The sgRNA oligos (designed with http://crispr.mit.edu/) were cloned into the pSpCas9(BB)-2A-Puro (PX459) (#48139; V2 #62988) vector. This was done according to the cloning strategy described in Ran *et al* (2013). For the generation of each enhancer deletion site, a flanking pair of sgRNA constructs was synthesized as DNA oligonucleotides (Eurofins, HPSF, no modifications). The top and bottom strands of sgRNA encoding oligonucleotide (final concentration of 100 µM) were annealed and phosphorylated by polynucleotide kinase (T4 PNK, Thermo Fisher scientific, #EK0031) reaction. The double-stranded and phosphorylated oligos were then diluted 1:8 and ligated into the pX459 vector using the restriction enzyme *Bbs*I (Bpil, 10 U/µl; Thermo Fisher Scientific, #FD1014) and the T4 DNA ligase (5 U/µl; Thermo Fisher Scientific, #EL0014). To digest any residual linearized DNA, a digestion with Plasmid-Safe exonuclease (10 U/µl; Biozym) was performed. Afterwards, the digested ligation reaction

was transformed into chemically competent *E. coli* bacteria. Successful cloning was validated by Sanger sequencing using a sequencing primer covering the *RNU6* promoter region 235 bp 5′ of the *Bbs*I site. Plasmids were transfected into the HeLa cells by the calcium-phosphate method. One day after transfection, cells were split 1:3 and puromycin (final concentration 1 µg/ml) was added to the medium until stable cell lines were established. All experiments were performed with pools of cells. The sequences of sgRNAs are listed in Appendix Table S1. These stable CRISPR cell lines were cultured in DMEM complete medium plus puromycin (1 µg/ml). To validate the engineered mutations at DNA level, cell pellets were recovered from 60-mm cell culture dishes and washed in PBS. Isolation of genomic DNA was performed using the NucleoSpin® Tissue kit (Macherey-Nagel) according to the manufacturer's instructions, and the DNA was eluted in 80 µl of elution buffer. Afterwards, the locus of interest was amplified by PCR using the GoTaq® Flexi DNA Polymerase (Promega) and the following PCR conditions: denaturation at 95°C for 2 min, 35 cycles of 95°C for 30 s, 60°C for 30 s, and 72°C for 40 s, and final elongation at 72°C for 5 min. The correct product size was analyzed on 1.5% agarose gels and the amplified DNA was isolated using the NucleoSpin® Gel and PCR Clean-up kit (Macherey-Nagel) according to the manufacturer's instructions. The DNA was eluted in $H_2O$. The isolated samples were prepared for sequencing according to the LGC guidelines for DNA. All primer pairs used for amplifying genomic DNA and sequencing are listed in Appendix Table S1. For HUVECs, sgRNAs were designed with the online CRISPR Design Tool (http://tools.genome-engineering.org) to target ~1-kbp regions around two enhancers in *BMP4* and *SAMD4A*. sgRNAs upstream of each enhancer were cloned into pSpCas9(BB)-2A-GFP (PX458) vector (Addgene plasmid #48138), while sgRNAs downstream of the enhancers were cloned into pSpCas9(BB)-2A-Puro (PX459) vector (Addgene, plasmid #48139). Human umbilical vein endothelial cells (HUVECs) were transfected via electroporation using 25 µg of each construct per $10^6$ cells in OptiMEM (20 ms pulse at 250 V in square waves on a Gene Pulser Xcell Electroporation System; Bio-Rad). After puromycin selection (1 µg/ml) for 48 h, cells were expanded for ~3 weeks. Genomic DNA was isolated and used as template in PCR and qPCR in order to validate and quantify the deletion. PCR products were sequenced to confirm the sequence of the sub-population carrying the enhancer's deletion.

## CRISPR–dCas9 activation (CRISPRa)

We used the structure-guided engineered CRISPR–dCas9 complex (Konermann *et al*, 2015) to mediate efficient transcriptional activation at endogenous genomic loci of *IL8* and *CXCL2*. The sequence-specific sgRNAs were designed following described guidelines, and sequences were selected to minimize off-target effects based on publicly available filtering tools (http://crispr.mit.edu/). The sgRNA oligonucleotides (produced by Eurofins) were cloned into lenti-sgRNA(MS2)-zeo backbone vector by Esp3I digestion. HeLa cells ($2.4 \times 10^5$ cells per 60-mm cell culture dish) were transfected by the calcium-phosphate method with a 1:1:1 mass ratio of lenti-sgRNA (MS2)-zeo sgRNA, lenti-dCAS9-VP64_Blast, and lenti-MS2-P65-HSF1_Hygro vectors (total plasmid mass of 12 µg/dish). Culture medium was changed 5 h after transfection. Twenty-four hours after transfection, cells were harvested for mRNA expression analysis by RT–qPCR.

## mRNA expression analysis by RT–qPCR

1 µg of total RNA was prepared by column purification (Macherey-Nagel or Qiagen) and transcribed into cDNA using Moloney murine leukemia virus reverse transcriptase (Thermo Fisher Scientific, #EP0352; or RevertAid Reverse Transcriptase, #EP0441) in a total volume of 20 or 10 µl. 2 or 1 µl of this reaction mixture was used to amplify cDNAs using assays on demand (0.25 or 0.5 µl) (Applied Biosystems/Thermo Fisher Scientific) for *ACTB* (Hs99999903_m1), *GUSB* (Hs99999908_m1), *IL6* (Hs00174131_m1), *IL8* (Hs00174103_m1), *NFKBIA* (Hs00153283_m1), *CXCL1 (*Hs00236937_m1), *CXCL2 (*Hs00236966_m1), *CXCL3 (*Hs00171061_m1), *CCL20 (*Hs00171125_m1), and *RELA (p65) (*Hs00153294_m1), as well as TaqMan Fast Universal PCR Master Mix (Applied Biosystems/ Thermo Fisher Scientific). Alternatively, primer pairs were designed and used with Fast SYBR Green PCR Master Mix (Applied Biosystems/Thermo Fisher Scientific). All PCRs were performed as duplicate reactions on an ABI 7500 real-time PCR instrument. The cycle threshold value ($c_t$) for each individual PCR product was calculated by the instrument's software, and the ct values obtained for inflammatory/target mRNAs were normalized by subtracting the ct values obtained for *GUSB* or *ACTB*. The resulting $\Delta c_t$ values were then used to calculate relative changes of mRNA expression as ratio (R) of mRNA expression of stimulated/unstimulated cells according to the following equation: $2^{-((\Delta ct\ stim.)-(\Delta ct\ unst.))}$.

## Cell lysis and immunoblotting

For whole-cell extracts, cells were lysed in Triton cell lysis buffer (10 mM Tris pH 7.05, 30 mM NaPPi, 50 mM NaCl, 1% Triton X-100, 2 mM $Na_3VO_4$, 50 mM NaF, 20 mM ß-glycerophosphate, and freshly added 0.5 mM PMSF, 2.5 µg/ml leupeptin, 1.0 µg/ml pepstatin, and 1 µM microcystin). Cell lysates or subcellular fractions were resolved in 7–12.5% SDS–PAGE gels, and immunoblotting was performed as described (Hoffmann *et al*, 2005). Separated proteins were electrophoretically transferred to PVDF membranes (Roth, Roti-PVDF 0.45 µm). After blocking with 5% dried milk in Tris–HCl-buffered saline/0.05% Tween (TBST) for 1 h, membranes were incubated for 12–24 h with primary antibodies, washed in TBST, and incubated for 1–2 h with the peroxidase-coupled secondary antibody. Proteins were detected by using enhanced chemiluminescence (ECL) systems from Millipore or GE Healthcare. Images were acquired and quantified using a Kodak Image Station 440 CF and the software Kodak 1D 3.6, or the ChemiDoc Touch Imaging System (Bio-Rad) and the software Image Lab V5.2.1 (Bio-Rad), or X-ray films and the software ImageJ.

## Co-immunoprecipitation

HeLa vector and Δp65$^{eIL8}$ cells were seeded in 145-mm cell culture dishes (3.5 × 10$^6$ cells), stimulated with IL-1α (10 ng/ml) for 0.5 and 1 h, and lysed in Triton cell lysis buffer. 15 µl of TrueBlot anti-rabbit IgG IP Beads (Rockland, # 00-8800-25) per sample was equilibrated in lysis buffer before adding 900 µl lysis buffer and 1 µg of primary antibodies (anti-NF-κB p65 sc-372 or normal rabbit IgG sc-2027). The samples were rotated for 2 h at 4°C and centrifuged at 2,500× *g* at 4°C for 1 min. The supernatant was removed and the pelleted beads were washed with 500 µl lysis buffer before

adding 750 µg of the cell lysates in a total volume of 900 µl lysis buffer. The samples were rotated for 2 h at 4°C, centrifuged at 2,500× *g* at 4°C for 1 min, and washed 3× with 900 µl lysis buffer with 5-min rotation steps at 4°C in between. After the last wash, the supernatant was aspirated and the beads were boiled in 60 µl 2× Roti-Load buffer (Carl Roth, #K929.1) for 10 min at 95°C. After spinning at 10,000 × *g* for 3 min, the supernatant was collected and 30 µl was loaded onto one SDS gel together with 50 µg of the simultaneously prepared cell lysates. Proteins were detected by immunoblotting using primary antibodies (anti-p65, anti-IκBα) followed by TrueBlot HRP-conjugated anti-rabbit IgG (Rockland, #18-8816-31).

## Puromycinylation assay

Parental HeLa cells or CRISPR-Cas9-based mutants were seeded in 10-cm cell culture dishes (1.4 × 10$^6$ cells). On the next day, cells were washed gently 4× with warm PBS and medium was replaced by 4 ml FBS-free medium with or without IL-1α (10 ng/ml) for 8 h. Thirty minutes prior to harvest, puromycin (10 µM) was added to the medium, and the supernatant was harvested and centrifuged at 15,000× *g* at 4°C for 30 min. Proteins from 1 ml of supernatant were precipitated by adding 1 ml of 11% TCA on ice for 45 min and centrifuged at 15,000× *g*/4°C for 15 min. The invisible pellet was washed with 1 ml of ice-cold 100% ethanol for 30 min at 4°C, centrifuged at 15,000× *g*/4°C for 15 min, shortly dried, and boiled in 50 µl 2× Roti-Load buffer. 10% of the samples were separated by a 12.5% SDS gel, and proteins were detected by silver staining. The remaining samples were analyzed by immunoblotting with an anti-puromycin antibody (Kerafast, EQ0001).

## Cytokine arrays

Human cytokine arrays were used for the analysis of secreted cytokines in cell culture supernatants. Parental HeLa cells or CRISPR-Cas9-based mutants were seeded in 60-mm cell culture dishes (5 × 10$^5$ cells). The following day, medium was replaced by 3 ml of complete medium (including FBS) for 8 h with or without IL-1α (10 ng/ml). Afterwards, the cell culture supernatant was harvested, centrifuged at 15,000× *g* at 4°C for 5 min, and stored at −80°C. The RayBio® C-Series Human Cytokine Antibody Array C5 (AAH-CYT-5-8) was performed with 1 ml of the thawed supernatant according to the manufacturer's instructions, including a sample incubation overnight at 4°C. Images were acquired and quantified using a ChemiDoc Touch Imaging System (Bio-Rad) and the software Image Lab V5.2.1 (Bio-Rad). Signal intensities of equally sized regions (defined by the volume of the largest spot) covering each arrayed spot were acquired using the volume tool of Image Lab V6.0.1 (Bio-Rad). The global background subtraction tool was used to obtain adjusted volume intensities (adjVI). These raw data are plotted in Appendix Fig S3 and were used for further calculations. Normalization was performed between two pairs of arrays (comparing untreated/IL-1α-treated) separately for vector controls and Δp65$^{eIL8}$ and Δ*RELA* cell lines. Mean signals from six positive controls (i.e., biotinylated antibody spots) of arrays performed with samples from untreated conditions (the reference array) were divided by the mean signals from positive controls of the IL-1α-treated samples to obtain the normalization factor (*n*). Fold changes

were calculated as follows: adjVI(IL-1α)*(n)/adjVI(untreated). The mean signals of all negative controls (no antibody spots) were used to define the threshold of detection.

## ELISA

Sandwich ELISAs from R&D Systems (DuoSet® ELISA human IL-8 (DY208) and IL-6 (DY206)) were used to measure secreted human IL-8 or IL-6 protein concentrations in cell culture supernatants from parental HeLa cells or mutant cell lines. The cells were seeded in 60-mm cell culture dishes ($5 \times 10^5$ cells), and at the next day the medium (3 ml) of all samples was exchanged. Then, cells were left untreated or were stimulated with IL-1α (10 ng/ml) for 3, 8, or 16 h in complete medium (including FBS). Thereafter, the cell culture supernatant was harvested, centrifuged at 15,000× *g* at 4°C for 15 s, and stored at −80°C. The samples were diluted in cell culture medium 1:5 (IL-8) or 1:20 (IL-6) and the ELISAs were performed according to the manufacturer's instructions using serial dilutions of recombinant IL-8 and IL-6 as standards. All samples were within the linear range of the standard curve. The obtained concentrations were normalized for cell number on the basis of total RNA concentrations obtained from the corresponding cell pellets.

## Immuno-RNA-fluorescence *in situ* hybridization (immuno-RNA FISH)

For detection of specific transcripts, the Affymetrix FISH kit QuantiGene® ViewRNA ISH Cell Assay (Life Technologies GmbH, QVC0001) was used in combination with specific branched-probe sets against *IL8* (VA4-13193, VA1-13103), *NFKBIA* (VA6-17971), *IL8*-intronic (VA1-6000437), *CXCL2*-intronic (VA6-6000438), and *ACTB* (VA4-10293) according to the manufacturer's instructions. For some experiments, this technique was further combined with classical indirect immunofluorescence. A total of 9,000 cells were seeded for 24 h in µ-slides VI (Ibidi) and washed twice with 1× PBS for 3 min. Subsequently, after fixation with 4% (w/v) paraformaldehyde in PBS (Santa Cruz, #281692) at 4°C overnight, cells were washed three times with 1× PBS for 3 min, permeabilized with the kit included detergent solution or PBS–Tween (1:1,000) at room temperature for 5 min and washed twice with 1× PBS for 3 min. For hybridization, probe sets (diluted 1:100) were incubated at 40°C for 3 h. The detection of labeled mRNAs was achieved using pre-amplifier mix, amplifier mix, and label probe mix (diluted 1:30, respectively), each incubated at 40°C for 30 min. Cells were washed twice for 2 min and once for 10 min with wash buffer. For combination with indirect immunofluorescence, the cells were washed for 1 min twice with 0.1% (w/v) saponin/Hanks' BSS (PAN, #P04-32505) and subsequently blocked with 10% (v/v) normal donkey serum in 0.1% (w/v) saponin/Hanks' BSS for 30 min. Protein detections were enabled by incubation with specific primary (anti-p65 F-6 mouse antibody, Santa Cruz, 8008) and secondary antibodies, incubated in 0.1% (w/v) saponin/Hanks' BSS at 37°C for 1 h. DyLight® 488-conjugated secondary antibodies (ImmunoReagent, DkxMu-003-D488NHSX) were diluted 1:100. Cells were washed three times for 10 min with 0.1% (w/v) saponin/Hanks' BSS or Hanks' BSS, respectively. As control, the primary antibody was omitted. Nuclei were stained with Hoechst 33342 (Invitrogen), and cells were finally embedded in 30% (v/v) glycerol/Hanks' BSS or Fluoromount-G

mounting medium (Thermo Fisher Scientific, #00-4958-02). For control of unspecific FISH signals, a "no FISH probe" Control (Ctl) was used for each experiment. This control excludes the FISH probe sets and includes pre-amplifier, amplifier, and label probes. In case of any unspecific signals, this control was used to determine the processing settings for the entire experiment. Fluorescence analyses were performed using the inverted microscope DMi8 (Leica) and the Leica LASX software (version 1.5.1.13187). Quantification of mRNA transcripts was performed using the Duolink Image Tool (version 1.0.1.2) from Olink Bioscience with the following settings for the signal channel red: nuclei size (px): 68; cytoplasm size (px): 100, signal threshold: 50–150, and signal size (px): 3–5.

## 3D-DNA FISH

3D-DNA FISH was performed using an adapted protocol based on Bolland *et al* (2013). Differentially labeled commercial DNA probes were obtained from Empire Genomics. Two large DNA probes mark *IL8* (IL8-20-RE, chr.4q13.3, 160 kb) and *IL6* (IL6-20-GR, chr.7p15.3, 194 kb) loci. Additionally, chromosome 4 control probes (CHR04-10-OR) were used, which bind to Chr.4p11 or Chr.4p13, respectively. A total of 9,000 (HeLa, Δp65$^{eIL8}$) to 12,000 cells (KB) were seeded for 24 h in µ-slides VI (Ibidi) and washed twice with PBS. Cells were fixed for 10 min at room temperature by using 4% (w/v) paraformaldehyde in PBS (Santa Cruz, #281692) and subsequently quenched with 0.1 M Tris–HCl (pH 7.4) for 10 min at room temperature. For permeabilization, cells were treated with 0.1% (w/v) saponin/0.1% Triton X-100/PBS for 10 min at room temperature, washed two times with PBS for 5 min, and incubated in 40% (v/v) glycerol in PBS for 3 h followed by 3 freeze-thaw cycles in liquid nitrogen. After thawing, cells were washed twice in PBS for 5 min, incubated in 0.1 M HCl for 30 min, washed with PBS for 5 min, permeabilized with 0.5% (w/v/v) saponin/0.5% Triton X-100/PBS for 30 min, washed again with PBS for 5 min, and pretreated with 50% (v/v) formamide/2× saline–sodium citrate (SSC), pH 7.0, for at least 30 min at room temperature. The hybridization mix contained 1.5 µl of each probe and 15 µl hybridization buffer (Empire Genomic) and was filled up to 50 µl with 50% (v/v/w) formamide/2× SSC/10% dextran sulfate (Sigma, #67578). Cellular DNA and probes were separately denatured at 75°C for 5 min. Probes were subsequently incubated on ice for 2 min and then prehybridized at 37°C for 10 min. Hybridization took place at 37°C in a humid chamber overnight. Samples were washed briefly with 2× SSC and then subjected to the following washing steps: 50% formamide/2× SSC for 15 min at 45°C, 1× SSC for 15 min at 63°C, 2× SSC for 5 min at 45°C, 2× SSC for 5 min, and PBS for 5 min at room temperature. Nuclei were stained with Hoechst 33342 (Invitrogen, 1 µM) for 5 min, and cells were finally embedded in 30% (w/v) glycerol/PBS. Fluorescence microscopy was performed using the inverted Leica DMi8 microscope and the Leica LASX software (version 1.5.1.13187). Z-stack images (in layers of 0.508 µm) were processed by 3D deconvolution, contrast changes, and background elimination.

## Immunofluorescence of LacO-array cells

Twenty-four hours prior to the experiment, cells were seeded on coverslips on 12-well plates. After treatment required for the specific

experiments, cells were washed twice with ice-cold PBS and fixed with 4% paraformaldehyde for 10 min at room temperature. After five washes with PBS, two of them for 5 min on a shaker, cells were permeabilized with 0.15% Triton/PBS for 15 min. Prior to first antibody incubation overnight at 4°C with gentle rocking, cells were blocked for 60 min at room temperature with 2% BSA in PBS with 0,05% Tween. The first antibody was prepared in the same solution. Afterward, cells were washed three times for 5 min with PBS in a shaker and the secondary antibody was added in the same solution as the first antibody and incubated for 2 h with gentle shaking. Cells were again rinsed briefly with PBS two times and washed three times for 5 min with PBS on a shaker. For staining of nuclei, Hoechst 33342 (1 µM in PBS) was added for 4 min, followed by two brief rinses with PBS and two washes with PBS for 5 min in a shaker. Cell-covered coverslips were then mounted on microscope slides using gelatin. Immunofluorescence images were acquired at room temperature with an Eclipse TE2000-E inverted microscope (Nikon) equipped with an X-Cite Series 120 fluorescence microscope light source (EXFO), a T-RCP remote control (Nikon), an ORCA-spark Digital CMOS camera C11440-36U (Hamamatsu), and a Nikon Plan Apo 100×/1.4 NA oil lens using NIS Elements AR 3.00 software (Nikon).

**Chromatin immunoprecipitation (ChIP) and data analysis**

Two 140-mm cell culture dishes were seeded with $3.7 \times 10^7$ HeLa cells, treated as described in the figure legends, and used for each condition. Proteins bound to DNA were cross-linked *in vivo* with 1% formaldehyde added directly to the medium. After 10 min of incubation at room temperature, 0.1 M glycine was added for 5 min to stop the cross-linking. Then, cells were collected by scraping and centrifugation at 1,610× *g* (5 min, 4°C), washed in cold PBS containing 1 mM PMSF, and centrifuged again at 1,610× *g* (5 min, 4°C). Cells were lysed for 10 min on ice in 3 ml ChIP lysis buffer (1% SDS, 10 mM EDTA, 50 mM Tris pH 8.1, 1 mM PMSF, Roche protease inhibitor mix). The DNA was sheared by sonication (28 × 30 s on/30 s off, power high; Bioruptor; Diagenode) at 4°C and lysates cleared by centrifugation at 16,100× *g* at 4°C for 15 min. Supernatants were collected and stored in aliquots at −80°C for subsequent ChIP. For determination of DNA concentration, 20 µl of sheared lysate was diluted with 100 µl TE buffer including 10 µg/ml RNase A. After 30 min at 37°C, 3.8 µl proteinase K (20 mg/ml) and 1% SDS were added and incubated for at least 2 h at 37°C followed by overnight incubation at 65°C. Samples were resuspended in two volumes of buffer NTB (Macherey-Nagel) and DNA was purified using NucleoSpin columns (Macherey-Nagel) according to the manufacturer's instructions. DNA was eluted with 50 µl 5 mM Tris pH 8.5, and concentration was determined by NanoDrop. For CHIP, the following antibodies were used: anti-histone H3 (2 µg; Abcam; ab1791), anti-NF-κB p65 (3 µg; Santa Cruz; sc-372), anti-phospho-Pol II (S5) (1.35 µg; Abcam; ab5131), H3K27ac (2 µg; Diagenode, pAb-174-050), H3K4me1 (2 µg; Abcam, ab8895), IgG (2 µg; Cell Signaling; 2729), and CTCF (4 µl; Millipore, 07-729). Antibodies were added to precleared lysate volumes equivalent to 25 µg of chromatin. Then, 900 µl of ChIP dilution buffer (0.01% SDS, 1.1% Triton X-100, 1.2 mM EDTA, 167 mM NaCl, 16.7 mM Tris–HCl pH 8.1) was added and the samples were rotated at 4°C overnight. Thereafter, 30 µl of a protein A/G-Sepharose mixture,

pre-equilibrated in ChIP dilution buffer, was added to the lysates and incubation continued for 2 h at 4°C. Beads were collected by centrifugation and washed once in 900 µl ChIP low-salt buffer (0.1% SDS, 1% Triton X-100, 2 mM EDTA, 20 mM Tris pH 8.1, 150 mM NaCl), once in 900 µl ChIP high-salt buffer (0.1% SDS, 1% Triton X-100, 2 mM EDTA, 20 mM Tris pH 8.1, 500 mM NaCl), once in 900 µl ChIP LiCl buffer (0.25 M LiCl, 1% NP-40, 1% desoxycholate, 1 mM EDTA, 10 mM Tris pH 8.1), and twice in 900 µl ChIP TE buffer (10 mM Tris pH 8.1, 1 mM EDTA) for 5 min at 4°C. Beads were finally resuspended in 100 µl TE buffer including RNase A (10 mg/ml). In parallel, 1/10 volume (2.5 µg) of the initial lysate (input samples) was diluted with 100 µl TE buffer including 10 µg/ml RNase A. After 30 min at 37°C, 3.8 µl proteinase K (20 mg/ml) and 1% SDS were added and both input and immunoprecipitates were incubated for at least 2 h at 37°C followed by overnight incubation at 65°C. Samples were resuspended in two volumes of buffer NTB (Macherey-Nagel) and DNA purified using NucleoSpin columns (Macherey-Nagel) according to the manufacturer's instructions. DNA was eluted with 50 µl 5 mM Tris pH 8.5 and stored at −20°C until further use. PCR products derived from ChIP were quantified by real-time PCR using the Fast ABI 7500 instrument (Applied Biosystems). The reaction mixture contained 2 µl of ChIP or input DNA (diluted 1:10 to represent 1% of input DNA), 0.25 µM of primers, and 10 µl of Fast SYBR Green Master Mix (2×) (Applied Biosystems) in a total volume of 20 µl. PCR cycles were as follows: 95°C (20 s) and 40× (95°C (3 s), 60°C (30 s)). Melting curve analysis revealed a single PCR product. Calculation of enrichment by immunoprecipitation relative to the signals obtained for 1% input DNA was performed. DNA isolated by CTCF-CHIP was subjected to NGS as described (Jurida *et al*, 2015). H3K27ac, H4K4me1, and p65 ChIP-seq data were analyzed as described previously (Jurida *et al*, 2015). Coverage vectors and peak sets from these data sets were visualized using R/Bioconductor package (Hahne & Ivanek, 2016). Quantitative comparison of binding signals was done by extracting read counts across all enhancer intervals and subsequent normalization of counts and detection of differentially bound regions by DESeq2 (Love *et al*, 2014). Normalized counts were plotted as bar plots for enhancer marked by significant de-regulation of H3K27ac binding signals within *CXCL2* and *CXCL8* regions, respectively.

**ATAC-seq and data analysis**

ATAC-seq was performed following a published protocol (Buenrostro *et al*, 2013). A total of 130,000 KB cells (untreated or treated with IL-1α for 1 h, 5Z-7-oxozeaenol (for 30 min) or combinations thereof) were harvested using trypsinization and washed with cold PBS. Cells were centrifuged in 50 µl cold PBS (500× *g*, 5 min, 4°C), and the supernatant was discarded. The cell pellet was resuspended in 50 µl cold lysis buffer (10 mM Tris–HCl pH 7.5, 10 mM NaCl, 3 mM $MgCl_2$, 0.1% IGEPAL CA-630), and the supernatant was discarded after another centrifugation step (500× *g*, 10 min, 4°C). The components for the transposase mix were part of the Illumina Nextera DNA library preparation commercial kit, and each cell pellet was incubated in 50 µl transposase mix (25 µl TD reaction buffer, 2.5 µl TDE1 Tn5 transposase, 22.5 µl nuclease-free water) at 37°C for 30 min (gentle shaking). The samples were then purified using the Qiagen MinElute PCR Purification commercial kit and

eluted in 10 μl elution buffer (10 mM Tris–HCl pH 8). For library preamplification, the purified 10-μl samples were amplified in thermocycler using 25 μl NEBNext High-Fidelity PCR Master Mix, 10 μl nuclease-free water, 2.5 μl Custom Nextera Primer 1 and 2 (25 μM), and the following program: 1 cycle of 72°C for 5 min and 98°C for 30 s, and 5 cycles of 98°C for 10 s, 63°C for 30 s, and 72°C for 1 min. 5 μl of this PCR mixed with 4.41 μl nuclease-free water, 0.25 μl Custom Nextera Primer 1 and 2 (25 μM), 0.09 μl 100× SYBR Green I, and 5 μl NEBNext High-Fidelity PCR Master Mix was used to determine the final amplification steps with real-time PCR (1 cycle of 98°C for 30 s, and 20 cycles of 98°C for 10 s, 63°C for 30 s, and 72°C for 1 min). Final library amplification was performed in the thermocycler with the remaining 45 μl library PCR mix with an additional cycle number calculated from real-time PCR cycle reaching 1/3 of the maximum fluorescence signal (see described program above for thermocycler). The amplified ATACSeq libraries were purified using the Qiagen MinElute PCR purification commercial kit and eluted in 20 μl elution buffer (10 mM Tris–HCl pH 8.0). Control of the libraries before sequencing was done by Agilent Bioanalyzer and additional test libraries separated with agarose gel electrophoresis (1% gel, 100 mV, 40 min). Following high-throughput sequencing on an Illumina platform, raw reads were mapped to the human genome (reference build hg19) using default BWA (default settings; Li & Durbin, 2010). Then, significant peaks (*q*-value $< 10^{-4}$ and $> 2$ fold enrichment over background) from each condition were selected and their genomic positions were used for further categorical analysis. The positions of peaks were intersected with H3K27ac ChIP-seq peaks (from Jurida *et al*, 2015), or raw ATAC-seq coverage of genomic regions from the different conditions was subjected to virtual footprinting for motif analysis by adapting the HINT subroutine of the Regulatory Genomics Toolbox suite (http://www.regulatory-genomics.org/). Coverage plots were generated using ngs.plot (Shen *et al*, 2014), while all other heatmaps were prepared using custom Python and R scripts (available on request).

## FAIRE assay

This assay was performed essentially as described (Rodriguez-Gil *et al*, 2018). A total of $1 \times 10^7$ cells were seeded in a T175 flask 24 h prior to the experiment. Subsequently, one flask was used per condition. The next day, cells were stimulated as required and cross-linked with formaldehyde (1% final concentration) for 10 min, followed by addition of glycine (100 mM final concentration) for 5 min. Cells were collected in the medium using a cell scraper and immediately put on ice. After centrifugation for 5 min at 4°C and 1,600 × *g*, the supernatant was aspirated and the pellet was resuspended in 2 ml ice-cold PBS with PMSF (0.5 mM final concentration) and again centrifuged. This step was repeated once. After final washes, the pellet was resuspended in 1 ml of ChIP lysis buffer (1% SDS, 10 mM EDTA, 50 mM Tris–HCl (pH 8.1); freshly added: 1 mM PMSF, 10 mM NaF, 0.5 mM sodium orthovanadate, 10 μg/ml leupeptin, 10 μg/ml aprotinin) and lysed for 10 min on ice. Sonication was carried out using a S220 Focused-ultrasonicator (Covaris) with 1-ml AFA tubes. Settings were as follows: peak: 150W; duty factor: 15; cycles per burst: 500 (during pause: 2.5W); and repetitions: 20. Sonicated lysates were then centrifuged at 16,100 × *g* for 15 min at 4°C and supernatants transferred to new reaction tubes. 50 μl of the lysates was kept at 4°C as cross-linked

sample. Another 50-μl aliquot of the same lysate was de-cross-linked as reference total DNA sample. For this purpose, 5 μl of RNase A (final concentration: 1 μg/μl) was added and sample was incubated for 30 min at 37°C. Next, 5 μl of proteinase K (final concentration: 2 μg/μl) was added and sample was incubated for 4 h at 37°C, then for 6 h at 65°C. For purification of DNA, the cross-linked as well as the de-cross-linked samples were diluted to 500 μl with ddH₂O. 500 μl of a phenol:chloroform:isoamyl alcohol mixture was added, and samples were vortexed before centrifugation at 15,800 × *g* for 10 min at 4°C. 450 μl of the aqueous upper part was transferred to a new reaction tube. After addition of 50 μl of NaCl (final concentration: 125 mM) and 450 μl of isopropanol, samples were mixed by inversion and incubated for 10 min at room temperature. After centrifugation at 15,800 × *g* for 10 min at 4°C, the supernatant was discarded and the pellet was washed with 200 μl of cold (4°C) 70% EtOH. The supernatant was again discarded and DNA pellets were left to air-dry before resuspension in 100 μl H₂O. For qPCR, 2 μl of the DNA was used per well as template. The de-cross-linked sample and the ACTB-TSS primer pair were used as reference to calculate the fold enrichment using the $\Delta\Delta C_T$ method with the following formula: $2^{-\Delta\Delta C_T} = ((C_T \text{ gene of interest} - C_T \text{ control gene}) \text{ cross-linked sample}) - ((C_T \text{ gene of interest} - C_T \text{ control gene}) \text{ de-cross-linked sample})$.

## Microarray transcriptomics and data analysis

The "Whole Human Genome Microarray 4x44K v2" (Agilent-026652, Agilent Technologies) covering the entire human transcriptome was used in this study. cRNA was transcribed from total RNA with the "Quick Amp Labeling kit, one-color" (#5190-0442; Agilent Technologies). cRNA synthesis, cRNA fragmentation, hybridization, and washing were carried out as recommended in the "One-Color Microarray-Based Gene Expression Analysis (Quick Amp Labeling)" guide (Agilent Technologies, G4140-90040 v5.7). Slides were scanned on an upgraded Agilent Microarray Scanner G2565 CA with a pixel resolution of 5 μm and a bit depth of 20. Data extraction was performed with the "Feature Extraction Software" V10.7.3.1 by using the default extraction protocol files "GE1_107_Sep09.xml".

Extracted data were imported as single-color experiments into GeneSpring GX V12.0 software (Agilent Technologies Inc., Santa Clara, CA). Data were log₂-transformed and quantile-normalized in GeneSpring GX. Low anti-log-transformed values were raised to 13, which was the mean value of all probes that were flagged by "Feature Extraction Software" as "not detected". In case of multiple Agilent probes for the same gene, the probe having the highest mean intensity was selected. A total of 21,765 probes on the array were assigned to an EntrezGeneID. Thereof, 14,204 genes were determined as expressed if they had an overall mean of at least 50 anti-log-transformed fluorescence units and were detected in at least 50% of the samples ("Feature Extraction Software" flag "detected"). In order to analyze the effect of IL-1α in vector controls, differentially expressed genes were identified using the moderated *t*-test of GeneSpring GX V12.0 software against the untreated vector controls. Ratio values with a *P*-value < 0.05 and > 3-fold changes were considered as significant changes. Differentially expressed genes in the three enhancer-deletion lines (Δp65^*eIL8*, Δp65^*eCXCL2*, and Δp65^*eIL8 + eCXCL2*) were calculated similarly compared to the untreated samples of each mutant using the moderated *t*-test

(ratio > 3-fold, $P < 0.05$). Overrepresentation analyses were performed on the web service Metascape (http://www.metascape. org, Zhou et al, 2019). Pathway & process enrichment was done against the 14,204 expressed genes as background set with the following settings: Min Overlap: 3, P-Value Cutoff: 0.01, Min Enrichment: 1.5, and selected sets: GO Molecular Functions, KEGG Functional Sets, GO Biological Processes, and KEGG Pathway.

### Intrinsic circular chromosome conformation capture (i4C), MC-i4C, and data analyses

i4C was performed on ~5 million wild-type or CRISPR-modified KB cells, HeLa cells, or HUVECs in two biological replicates (unless stated otherwise). Preparation of i4C templates, using *Apo*I as the primary restriction enzyme and the promoter/enhancer regions of *IL8* and *CXCL2* as viewpoints, was performed exactly as described by Brant et al (2016), while PCR-based generation of i4C libraries for sequencing on a HiSeq 2500 platform (Illumina) was as previously described (Stadhouders et al, 2013) using the primers listed in the Key Resources Table. Following sequencing to ≥ 8 million reads, analysis was performed using the *foursig* algorithm (Williams et al, 2014) to obtain a catalogue of significant *cis*-interactions for each replicate and viewpoint; significant interactions shared by independent replicates (where applicable) are presented under each i4C signal track. From the same data, all *trans*-interactions were examined and intersected manually. For the multi-contact (MC) i4C data from HUVECs, we generated i4C libraries exactly the same way, PCR-amplified by the primers listed in Appendix Table S1, and subjected to long-read sequencing on a PacBio® platform (via BaseClear, NL). Data analysis was performed as described in Allahyar et al (2018) starting from CCS reads (to which raw data were converted, provided directly by BaseClear), and all interaction data were finally presented after assigning to non-overlapping 10-kbp bins along chromosome 14 and visualized using custom R scripts (available upon request).

### Quantification and statistical analysis

Bands detected by immunoblotting were quantified using ImageJ (http://rsbweb.nih.gov/ij/) or Bio-Rad Image Lab, version 5.2.1 build 11. Statistics (Mann–Whitney rank-sum test, t-tests, Fisher's exact test) were calculated using SigmaPlot 11, GraphPad Prism 6.0, or Microsoft Excel 2013, or online at https://www.graphpad.com/ quickcalcs. Unless stated otherwise, in all box plots the boundary of the box closest to zero indicates the 25th percentile, a black line within the box marks the median, a red line marks the mean, and the boundary of the box farthest from zero indicates the 75th percentile. Whiskers (error bars) above and below the box indicate the 90th and 10th percentiles. Points mark the remaining outliers.

## Data availability

The following published NGS data sets were used. For KB cells, microarray, RNA-seq, and ChIP-seq data are available via our previous NCBI GEO submissions with the accession numbers GSE64224 and GSE52470 (https://www.ncbi.nlm.nih.gov/geo) (Jurida et al, 2015). For HeLa-S3 cells and HUVECs, all ChIP-seq data are from the ENCODE project (www.encodeproject.org). New ATAC-seq, CTCF ChIP-seq, and microarray data generated here are available via the NCBI GEO repository under the accession number GSE134436 (https://www.ncbi.nlm.nih.gov/geo), and all i4C data are available via the SRA repository under the accession number PRJNA552438 (https://www.ncbi.nlm.nih.gov/bioproject).

Bed files, bed graph files, and scripts for all i4C data are available as compressed source data for Expanded View and Appendix figures.

### Other resources

A detailed list of reagents, antibodies, and oligonucleotide sequences used in this study is provided in Appendix Table S1.

**Expanded View** for this article is available online.

## Acknowledgements

This work was supported by grants from the Deutsche Forschungsgemeinschaft KR1143/7-3 (project 162103480), KR1143/9-1 (KLIFO309, P3, project 284237345), TRR81/2 (B02, project 183582903), SFB1213 (B03, project 268555672) and SFB1021 (C02, project 197785619) (to M.K.), SCHM 1417/8-3 (project 162103480), TRR81/2 (A07, project 183582903), SFB1021 (C01, project 197785619) (to M.L.S.), and PA2456/4-1 (project 285697699) and PA2456/5-1 (project 290613333) (to A.P.). Work in the M.L.S. laboratory is also supported by the Deutsche Krebshilfe (111447) and the IMPRS program of the Max-Planck Society; work in both M.K. and M.L.S. laboratories is further supported by the Excellence Cluster Cardio-Pulmonary System and Cardio-Pulmonary institute (EXC 147: Kardiopulmonales System, project 24676099; EXC 2026: Cardio-Pulmonary Institute (CPI), project 390649896) and the DZL/UGMLC program; and work in the A.P. laboratory is also supported by core funding by the Center for Molecular Medicine Cologne (ZMMK). We wish to also thank Amin Allahyar and Wouter de Laat, Utrecht, for help with MC-i4C analysis.

### Author contributions

S-SW generated the enhancer/promoter deletions and extensively characterized the cell lines; JM-S performed CRISPRa experiments and all experiments for the revisions together with HW; HW performed mRNA stability assays; HM performed immunoblots; CM-B and KB contributed to 2D-/3D-FISH analyses; LJ performed ATAC-seq experiments; AB performed 3D-DNA FISH; UT established puromycinylation assays and vector control lines; AN and TS performed NGS analyses; OD-B and AW performed and analyzed microarray experiments; AM and LB generated i4C data; AM generated and analyzed HUVEC enhancer-deletion lines; WFJI sequenced i4C libraries; TG analyzed i4C and ATAC-seq data; TR and MLS conceived, performed, and analyzed LacO-LacI-p65 and FAIRE experiments; AP and MK conceived and supervised the study and analyzed and visualized data; S-SW, JM-S, CM-B, AB, KB, MB, and TG analyzed data and together with AP, MLS, and MK finalized the manuscript. All authors approved the final version of the manuscript.

### Conflict of interest

The authors declare that they have no conflict of interest.

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
