## [Review Process File · The EMBO Journal]

Distinct IL-1 α -responsive enhancers promote acute and coordinated changes in chromatin topology in a hierarchical manner

Sinah-Sophia Weiterer, Johanna Meier-Soelch, Theodore Georgomanolis, Athanasia Mizi, Anna Beyerlein, Hendrik Weiser, Liliya Brant, Christin Mayr-Buro, Liane Jurida, Knut Beuerlein, Helmut Müller, Axel Weber, Ulas Tenekeci, Oliver Dittrich-Breiholz, Marek Bartkuhn, Andrea Nist, Thorsten Stiewe, Wilfred F.J. van IJcken, Tabea Riedlinger, M. Lienhard Schmitz, Argyris Papantonis and Michael Kracht

Review timeline:

Submission date:	11th Jan 2019
Editorial Decision:	27th Feb 2019
Revision received:	29th Jul 2019
Editorial Decision:	9th Sep 2019
Revision received:	27th Sep 2019
Accepted:	1st Oct 2019

Editor: Karin Dumstrei

Transaction Report:

1st Editorial Decision

27th Feb 2019

Thank you for submitting your manuscript to The EMBO Journal. Your study has now been seen by two referees and their comments are provided below.

As you can see from the comments, both referees find the analysis interesting. They raise a number of relevant issues that I would like to ask you to address in a revised version. Please also make sure that you make the data available to the referees when you submit the revised version.

REFeree REPORTS:

Referee #1:

In this manuscript Weiterer et al. examined the mechanisms of IL-1 induced chromatin topology changes at the IL8/CXCL2 and BMP4/SAMD4A loci. They have used different molecular methods, including ATAC-seq, FAIRE-qPCR, CRISPR, ChIP, Immuno-RNA FISH, LacI-LacO reporter systems, Microarrays and i4C-seq, to describe how a hierarchical crosstalk between two cytokine-activated enhancers, leads to the formation of higher-order hubs that control the chemokine locus expressing CXCL1-3 and IL8 (CXCL8) in human epithelial cells.

Technically, the study, is well done, thorough, convincing and supports the importance of chromatin-based mechanism for the regulation of genes involved in the inflammatory response.

However, the manuscript is relatively confusing and difficult to read including unnecessary data that do not appear to be required to support the main theme of the work. For example, I don't understand

the necessity of Figures 1 and 2. Importantly, the information provided at the figure legends is absolutely minimal to guide the reader to understand the figures.

In summary, I believe that the paper contains important new information and it should be extensively revised to streamline its message with main focus on the study of the role of the two minimal NF- κ B bound enhancers and their role in coordinating the expression of IL-1 expression via topological chromatin changes.

Comments:

1. Figure S4. The authors should provide an explanation for the inability of NF- κ B to translocate into the nucleus. The deletions of the IL8-CXCL2 enhancers lead to the reduction of the NF- κ B nuclear translocation. It would be useful to check the p65 protein levels alongside with the I κ B α protein levels in the cell lines mentioned in Figure S4. Furthermore, a co-IP experiment (IP against p65 and detection of I κ B α via WB) could also show if there is any difference in the bound-release of p65-I κ B α upon IL1-a among these cell lines. Finally, it would be useful to add the RELA mRNA FISH and the RELA transcription/cell bar plots.
2. Figure 6B. The authors should show the impact of the IL1-a stimulation on the gene expression of the chemokine locus' genes in the Δ RELA, Δ p65eIL8, Δ p65pIL8 cell lines.
3. Figure 3A. ChIP-seq data, H3K27ac +IL-1a,+TAKi. As expected, there is a reduction of the H3K27ac occupancy on the region of the IL8 promoter/enhancer after the incubation of IL-1a and TAKi compared to the incubation of IL-1a alone. On the other hand, there is an enrichment of the binding events on the CXCL2 enhancer and there is no difference on the CXCL2 promoter. Can the authors explain the opposite result of the binding of p65 in CXCL2 locus as compared to its binding at the IL8 locus?
4. Figure S5C. α -I κ B α . In the empty vector samples, after the IL-1a stimulation, there is a reduction on the I κ B α protein levels. In the same samples of Figure S5D, there is no difference between the stimulated and the unstimulated I κ B α protein levels.
5. Figure S2B. A box plot for the chemokine locus' genes should be added in order to emphasize not in IL1-a regulated genes in general but specially in those located in IL8-CXCL2 locus.
6. The NF- κ B and the RNAPII ChIP-seqs that are mentioned in Figure 3B, should be added also in Figure 3A or at least as Supplementary Figure for-CXCL2 locus.

Minor comments:

1. Page 8, line 7. There is a reference in Figure 3C that does not exist. There is only Figure 3A and B.
2. Page 16, lines 3-5. There is no RNA polymerase II ChIP-seq and H3K4me3 data as mentioned in this Figure legend.
3. Figure S3B, H3K4me1, CXCL3 promoter. There are no values-line for the empty vector sample.
4. Figures S8-S9. All the i4C profiles do not have the "contact length" information that was provided in all the other similar Figures of the paper e.g. Figure3,4 and S7.
5. Page 16, Figure legend 5, line 8. Word "und" has to be corrected.

Referee #2:

The manuscript by Weiterer et al is an interesting study reporting an in-depth characterization of enhancer function within the human CXCL/IL8 inflammatory gene locus. While the effects of master enhancer deletion are quite striking, the study would be improved by some attempt to determine the basis of the observed broader effects of enhancer deletion and also appropriate validation of the cells systems used. The following points should be addressed before considering the paper for publication in EMBO J.

Main points:

- 1) A key question left unanswered is how the IL8 enhancer functions beyond the CXCL/IL8 TAD to promote expression of almost all IL-1 induced genes. It would have been more informative if the PacBio methods used for BMP4/SAMD4A at the end of paper had been done with the IL8 enhancer KO cells. The authors don't attempt to explain further why the IL8 enhancer seems to affect all IL-1a induced genes. Could it be related to the CXCL locus being a critical early response locus and that its dysregulation has consequences for the rest of the gene program? How might this occur? Is

the defective NFkB translocation in KO cells suggesting that this locus helps to maintain NFkB nuclear occupancy for an extended period to support expression from further distal loci?

2) The authors have employed KB cells for the ATAC-seq and i4C experiments. This cell line has been at the center of a longstanding controversy over misidentification. It was originally reported by Harry Eagle to be derived from a laryngeal squamous cell carcinoma, but was later found to actually correspond to HeLa cells. Even the original stock of KB deposited by Eagle at ATCC was eventually found to be of HeLa cell origin. This cell line has been repeatedly used to exemplify the dangers of misrepresentation and the importance of authenticating cell lines in biomedical research. The authors seem to treat KB and HeLa cells as different in the manuscript, but the "Cell lines and cytokine treatments" section of the methods does not mention the origin of the KB cells that were used, how this specific stock is different from HeLa cells, or what efforts (if any) were made to authenticate each of the cell lines that were used.

3) To better justify the use and comparison KB and HeLa cell lines, the authors could conduct additional analysis of their own data, as similar i4C experiments were done in both cell lines. To what extent do these profiles compare? The authors should also compare their ChIP seq profiles for KB cells (Fig 3A) from this paper with HeLa from ENCODE (Fig S1) to determine how comparable the two cell lines are at the chromatin level.

4) While HeLa cells and other cancer cell lines have been a valuable tool in several fields, their genomic composition tends to differ so much from that of primary cells that it is reasonable to question their utility for functional genomics studies. In the specific case of HeLa cells, aneuploidy, large structural variants, and even chromothripsis have been well documented. The manuscript does not provide evidence that the main CXCL cluster in 4q, in which much of the work is centered, is not affected by large structural variants in the cells they have studied. In the absence of experimental evidence, this serious limitation of cancer cell lines should at least be presented and discussed, as it affects the generalizability of the findings reported.

5) In the intronic RNA FISH experiments, the authors report that either of the two enhancer microdeletions introduced by CRISPR led to reduced CXCL2 and IL8 transcription after IL-1 stimulation, without evidence of the hierarchical relationship between the two enhancers that has been suggested by the i4C experiments. They then make an attempt at stratifying the signal by allele-specific patterns of expression. The way in which this was done needs to be more clearly explained. As currently written, it is unclear to what extent the co-localization of signals from CXCL2 and IL8 provides a reliable and quantitative method for measuring allele-specific expression. Unless this can be shown convincingly, an alternative method that involves measurement of expressed sequence variants would be necessary to draw meaningful conclusions about differential effects of the two enhancer microdeletions.

6) The effects of enhancer knockout on genes both within and beyond the CXCL locus is striking. From an inflammation regulation perspective, it would be important to confirm that the effects of diminished gene induction in enhancer deleted cells are reflected in secreted cytokine levels. The cytokines both within (IL8, CXCL1, 2 and 3) and outwith the locus (IL6, CCL20) could be readily measured with commercially available detection kits.

7) The authors should comment on why the enhancer KO cells appear to show substantially increased i4C contacts compared to controls (Fig4 IL8 enhancer plots). There is a very brief mention that the CXCL2 enhancer deleted cells have 'rich interactomes', and no mention for the IL8 enhancer, and it is not noted for either case that the contact numbers actually go up. How would the authors explain master enhancer deletions causing increased levels of chromatin contacts within the TAD?

8) The paper is severely lacking in terms of data access and availability. I don't consider it reasonable practice to state that data will be made available 'upon paper acceptance', as it prevents appropriate peer review of the manuscript data presentation and transparency.

Minor points:

1) Fig 1B would be more easily interpreted as a 3-way Venn diagram. This would determine how

similar the no IL-1 condition is to IL-1 treated cells in presence of the TAK1 inhibitor.

2) Fig 2 is not so central to the paper and should be a supplementary figure. Similarly, Fig S9 is more important and would be better as a main figure.

1st Revision - authors' response

29th Jul 2019

Please see next page.

Thank you very much for inviting us to revise our manuscript entitled “**Hierarchical IL-1 α -responsive enhancers act via acute and coordinated changes in chromatin topology**” and also for granting us an extension for this resubmission. Meanwhile, we have thoroughly addressed all concerns raised by the reviewers by both, several additional experiments and by restructuring and rewriting the manuscript along the lines suggested. All changes to the text are colored in red.

We believe that the reviewers’ suggestions and the new data substantially strengthen our previous conclusions and provide further support for chemokine enhancers acting as a new type of master regulator of inflammatory gene expression patterns. Specifically, we have now addressed the issues raised as follows:

Referee #1:

In this manuscript Weiterer et al. examined the mechanisms of IL-1 induced chromatin topology changes at the IL8/CXCL2 and BMP4/SAMD4A loci. They have used different molecular methods, including ATAC-seq, FAIRE-qPCR, CRISPR, ChIP, Immuno-RNA FISH, LacI-LacO reporter systems, Microarrays and i4C-seq, to describe how a hierarchical crosstalk between two cytokine-activated enhancers, leads to the formation of higher-order hubs that control the chemokine locus expressing CXCL1-3 and IL8 (CXCL8) in human epithelial cells. Technically, the study, is well done, thorough, convincing and supports the importance of chromatin-based mechanism for the regulation of genes involved in the inflammatory response. However, the manuscript is relatively confusing and difficult to read including unnecessary data that do not appear to be required to support the main theme of the work. For example, I don’t understand the necessity of Figures 1 and 2. Importantly, the information provided at the figure legends is absolutely minimal to guide the reader to understand the figures. In summary, I believe that the paper contains important new information and it should be extensively revised to streamline its message with main focus on the study of the role of the two minimal NF-kB bound enhancers and their role in coordinating the expression of IL-1 expression via topological chromatin changes.

Reply: We thank this reviewer for his fair and positive evaluation of our work and for his/her valuable criticisms. We found all concerns valid and have addressed them comprehensively by additional experiments and by rearranging and/or rephrasing the manuscript including providing additional information in the legends. In the course of these amendments, we have opted to keep **Fig. 1** as a main figure. We feel that these ATAC-seq data and their analysis are important for the cytokine field and that they provide a nice introduction to IL-1-mediated genome-wide changes in chromatin accessibility (possibly also underlying changes in spatial chromatin structure). However, according to the suggestions, we have now moved **Fig. 2** to the supplement (now **Fig. EV1**).

Comments:

1. Figure S4. The authors should provide an explanation for the inability of NF- κ B to translocate into the nucleus. The deletions of the IL8-CXCL2 enhancers lead to the reduction of the NF- κ B nuclear translocation. It would be useful to check the p65 protein levels alongside with the I κ B α protein levels in the cell lines mentioned in Figure S4. Furthermore, a co-IP experiment (IP against p65 and detection of I κ B α via WB) could also show if there is any difference in the bound-release of p65-I κ B α upon IL1-a among these cell lines. Finally, it would be useful to add the RELA mRNA FISH and the RELA transcription/cell bar plots.

Reply: We have now added immunoblot experiments showing that basal I κ B α and p65 protein levels remain unchanged between the various enhancer mutant cells compared to vector controls (new **Fig. EV8**). In contrast, the IL-1-induced I κ B α degradation is partially suppressed in the enhancer mutant cell lines, in particular in $\Delta p65^{eIL8}$ cells (new **Fig. EV8A, B**). We also performed the Co-IP experiment which is consistent with reduced activation of the canonical NF- κ B pathway in the $\Delta p65^{eIL8}$ cell line as it shows more I κ B α remaining bound to p65 at the peak of IL-1 stimulation at 30 min (new **Fig. EV8D**). Finally, we added RELA (p65) mRNA expression values in $\Delta p65^{eIL8}$ and $\Delta p65^{eCXCL2}$ (**Fig. EV8C**) and $\Delta RELA$ (**Fig. EV9D**) cells. These results are consistent with our protein level analysis, showing no change in p65 levels. Therefore, we have decided to not go ahead with additional FISH experiments to detect p65 mRNA. We conclude that the mutant IL8 enhancer reduces signaling via the canonical NF- κ B pathway, which can be placed downstream of the IL-1 receptor complex (see below) and upstream of the IKK complex as we observe a reduction in P-I κ B α / I κ B α and a concomitant reduced degradation of I κ B α (new **Fig. EV8A, B**). The increased cytosolic levels of I κ B α explain the diminished nuclear translocation of p65 shown in **Fig. EV7** (previous Fig. S4). Most likely autocrine or paracrine signaling is missing that feeds into the IL-1-mediated activation of the canonical NF- κ B pathway via IL-8 or a related secreted factor. Alternatively, one could suggest that some (unknown) negative regulators are upregulated upon enhancer mutation. While the mechanistic elucidation of these defects is beyond the scope of the current study, we provide comments and ideas on these issues in the Discussion section of the manuscript.

2. Figure 6B. The authors should show the impact of the IL1-a stimulation on the gene expression of the chemokine locus' genes in the $\Delta RELA$, $\Delta p65^{eIL8}$, $\Delta p65^{pIL8}$ cell lines.

Reply: For the $\Delta p65^{eIL8}$ and $\Delta p65^{pIL8}$ lines, these data were already included in previous Figs S1B and S5E. They are now shown in **Figs 3B** (for $\Delta p65^{eIL8}$ and $\Delta p65^{eCXCL}$) and **EV9C** (for $\Delta p65^{pIL8}$). For $\Delta RELA$, we now added additional mRNA expression data as a new **Fig. EV9D**.

3. Figure 3A. ChIP-seq data, H3K27ac +IL-1a,+TAKi. As expected, there is a reduction of the H3K27ac occupancy on the region of the IL8 promoter/enhancer after the incubation of IL-1a and TAKi compared to the incubation of IL-1a alone. On the other hand, there is an enrichment of the binding events on the CXCL2 enhancer and there is no difference on the CXCL2 promoter. Can the authors explain the opposite result of the binding of p65 in CXCL2 locus as compared to its binding at the IL8 locus?

Reply: The CXCL2 locus shows wide-spread H3K27acetylation that is illustrated by fairly compressed panels shown in **Fig. 2A** (previous Fig. 3A). To better illustrate the effects of TAK1 inhibition, we have prepared a new supplementary figure with a zoomed-in view for the IL8 and CXCL2 enhancer/promoter regions from our published ChIP-seq data sets, also showing quantification of H3K27ac H3K4me1 and p65 read counts, before and after IL-1 treatment and in the presence/absence of TAK1 inhibition (new **Fig. EV2**). Together, this data demonstrate the specific effect of TAK1 inhibition on these p65-bound enhancers regions. Additional ChIP-qPCR data showing the effect of TAKi or TAK1 depletion on H3K27ac support this conclusion, also for HeLa cells, and have been previously published in our Jurida et al. paper (Jurida, Soelch et al., 2015).

4. Figure S5C. α -I κ B α . In the empty vector samples, after the IL-1a stimulation, there is a reduction on the I κ B α protein levels. In the same samples of Figure S5D, there is no difference between the stimulated and the unstimulated I κ B α protein levels.

Reply: We have repeated the experiments and have run the samples side by side on the same gels. This new data show the prototypical IL-1-mediated reduction of I κ B α after 30 min (new **Fig. EV9B**). This same effect is also seen in "empty vector" control samples from other experiments (new **Fig. EV8A,B**).

5. Figure S2B. A box plot for the chemokine locus' genes should be added in order to emphasize not in IL1-a regulated genes in general but specially in those located in IL8-CXCL2 locus.

Reply: We did this analysis using the microarray data shown in **Fig. 3C and Fig. EV4C-D** (previous Fig. S2B). Chromosome 4 contains 813 annotated genes of which 481 were expressed above background in HeLa cells. The only other chemokine gene expressed in the TAD we study here (besides IL8, CXCL1, CXCL2, CXCL3) is CXCL5, which is

however only very weakly (1,6-fold) regulated by IL-1. Thus, of the 15 annotated genes present in this TAD, only 8 were found to be expressed, three of which are not chemokines (see Fig.1 for review).

Fig. 1 (for review). Expression pattern of 15 genes contained in the chemokine TAD located on chromosome 4 (74439028-75320577).

(A) Summary of all 15 genes.

(B) Expression pattern using normalized fluorescence intensity (FI) values from the microarray experiments shown in Table EV1. The gray line marks background signals, above which mRNAs were considered to be significantly expressed.

Given this low number of additional genes, we felt that these might not be that informative and instead decided to add box plots for all 481 expressed genes in chr. 4 (new Fig. 3C), which also addresses the concerns of the other reviewer #2 (see point 4, below) on the integrity of the entire chromosome itself. These 481 genes are expressed as expected across all mutant enhancer lines, arguing against any major detrimental effect of the microdeletions to overall structure of chr. 4.

6. The NF- κ B and the RNAPII ChIP-seqs that are mentioned in Figure 3B, should be added also in Figure 3A or at least as Supplementary Figure for-CXCL2 locus.

Reply: We find that this figure is already rather busy, while the ChIP-seq profiles and data referred to in Fig. 2B have also been extensively presented in our previous works (see Jurida et al., 2015). The RNA PolII data may be viewed in this paper and downloaded as .bw files to be viewed in a genome browser (GEO accessions: GSE64224 and SE52470). As mentioned above (reply to comment 3), we nonetheless prepared a new Fig. EV2 showing p65 ChIP-seq profiles.

Minor comments:

1. Page 8, line 7. There is a reference in Figure 3C that does not exist. There is only Figure 3A and B.

Reply: We have corrected this error.

2. Page 16, lines 3-5. There is no RNA polymerase II ChIP-seq and H3K4me3 data as mentioned in this Figure legend.

Reply: We have corrected these errors.

3. Figure S3B, H3K4me1, CXCL3 promoter. There are no values-line for the empty vector sample.

Reply: This was a mistake that has been corrected. The figure is now Fig. EV5B.

4. Figures S8-S9. All the i4C profiles do not have the "contact length" information that was provided in all the other similar Figures of the paper e.g. Figure 3,4 and S7.

Reply: we assume that the reviewer refers to the “contact strength” colour key (length bars are provided in all i4C figures), and the reason that these two figures do not include one, is the lack of true replicates. We have performed this statistical analysis using “pseudoreplicates”, but feel that this is not meaningful to present, and leave the i4C profiles in these two figures to be qualitatively compared.

5. Page 16, Figure legend 5, line 8. Word "und" has to be corrected.

Reply: the word “und” has been changed to “and” (now legend of **Fig. 5**).

Referee #2:

The manuscript by Weiterer et al is an interesting study reporting an in-depth characterization of enhancer function within the human *CXCL/IL8* inflammatory gene locus. While the effects of master enhancer deletion are quite striking, the study would be improved by some attempt to determine the basis of the observed broader effects of enhancer deletion and also appropriate validation of the cells systems used. The following points should be addressed before considering the paper for publication in *EMBO J*.

Reply: We thank this reviewer for his/her positive and thorough evaluation. We found all concerns risen below to be valid, and have addressed them by additional experimentation and by amending the manuscript text accordingly.

Main points:

1) A key question left unanswered is how the *IL8* enhancer functions beyond the *CXCL/IL8* TAD to promote expression of almost all *IL-1* induced genes. It would have been more informative if the PacBio methods used for *BMP4/SAMD4A* at the end of paper had been done with the *IL8* enhancer KO cells. The authors don't attempt to explain further why the *IL8* enhancer seems to affect all *IL-1a* induced genes. Could it be related to the *CXCL* locus being a critical early response locus and that its dysregulation has consequences for the rest of the gene program? How might this occur? Is the defective NF κ B translocation in KO cells suggesting that this locus helps to maintain NF κ B nuclear occupancy for an extended period to support expression from further distal loci?

Reply: The point the reviewer raises is certainly valid, and was also touched upon by Reviewer #1 (see above). We agree with the interpretation that the *IL8/CXCL2* locus has some type of priming function for the entire *IL-1* response. This effect is seen at different levels of signaling and mRNA expression and we provide now additional data (shown in new **Figs 3, EV6, and EV8**) and a more elaborate discussion on how it is most likely that a paracrine/autocrine *IL-8*, *IL-6*, *MIP-3 α /CCL20* (or related) dependent signal is missing that is apparently required for sustained activation of the canonical NF- κ B pathway. However, all this is not due to changes in the mRNA or protein levels of NF- κ B, but we do record a small yet sustained reduction in I κ B α degradation that can explain the overall inefficient translocation of NF- κ B into cell nuclei, and thus the dampening of the *IL-1* response. There is also a limited number of papers that have observed *IL-8* autocrine effects including activation of TRAF6-NF- κ B (e.g.,(Manna & Ramesh, 2005)). We cite these additional references and comment on this issue in the Discussion section of the revised manuscript.

Finally, as regards the PacBio-i4C, we agree that its application on the *IL8/CXCL2* TAD is likely to yield interesting data, but this is beyond our reach right now for a number of reasons. First, this method is still very new and troubleshooting on a locus-to-locus basis has proved rather time-consuming; second, it is a very expensive approach (sequencing of just 3 libraries costs >7500 EUR); third, we rely on people outside our laboratories for data analysis that proved a bottleneck for the analysis of the existing data anyway. As a result, this would be unfeasible and not affordable at this moment, and we hope that in the near future cheaper approaches might render this easier.

2) The authors have employed KB cells for the ATAC-seq and i4C experiments. This cell line has been at the center of a longstanding controversy over misidentification. It was originally reported by Harry Eagle to be derived from a laryngeal squamous cell carcinoma, but was later found to actually correspond to HeLa cells. Even the original stock of KB deposited by Eagle at ATCC was eventually found to be of HeLa cell origin. This cell line has been repeatedly used to exemplify the dangers of misrepresentation and the importance of authenticating cell lines in biomedical research. The authors seem to treat KB and HeLa cells as different in the manuscript, but the "Cell lines and cytokine treatments" section of the methods does not mention the origin of the KB cells that were used, how this specific stock is different from HeLa cells, or what efforts (if any) were made to authenticate each of the cell lines that were used.

Reply: This is a fair and justified criticism. KB and HeLa cells have been utilized for more than two decades in the *IL-1* field by various labs (e.g., (Bird, Sleath et al., 1991, Saklatvala, Kaur et al., 1991)). We have been using them for a long time as a cell culture model of transformed epithelial cell lines, and in this context it is irrelevant from which tumor (or individual) they were originally derived. Both are well suited for studies on *IL-1* signaling, because, like many cell lines derived from solid tumors that we have been using in the lab, they express all relevant components of

the IL-1 system and display a fully functional IL-1 response. In fact, in our experience there is no principal difference between IL-1 signaling in diploid epithelial cells (such as RPE-1; also used for comparison in this study) or HeLa or KB, respectively. We are fully aware of the (critical) literature on KB cells and their identification as HeLa cells, which is mainly based on satellite markers and other genetic tests. However, whereas the HeLa genome has been recently sequenced to substantial depth (Adey, Burton et al., 2013, Landry, Pyl et al., 2013), no such effort is, to our knowledge, available for KB cells. In our hands, the batch of KB cells we are using (originally obtained from J. Saklatvalas' laboratory in Cambridge, UK) remain consistently morphologically different to HeLa, and show a more homogenous IL-1 response at the single-cell level, as judged by immuno-RNA FISH (see **Fig. EV3A-C**). They also differ in copy numbers of chr. 4 (see **Fig. EV3D**). Nonetheless, to conclusively resolve all these issues, we decided to send genomic DNA from our HeLa and KB stocks for genetic testing by the DSMZ-German Collection of Microorganisms and Cell Cultures facilities (<https://www.dsmz.de/dsmz>). The results show that they are 100% identical to the original HeLa cells deposited at DSMZ and ATCC (see **Fig. EV2F**). We thus conclude that these KB cells are a HeLa derivative with some phenotypic differences. To handle these issues as transparently as possible, and to provide this information for all labs using these cell lines as models, we have decided to include all these data as **Fig. EV3** and have also added an extended paragraph to the results section. Importantly though, these lines have different copy numbers of chromosomes, and this in-depth characterization does not at all affect the conclusions drawn from our experiments.

3) To better justify the use and comparison KB and HeLa cell lines, the authors could conduct additional analysis of their own data, as similar i4C experiments were done in both cell lines. To what extent do these profiles compare? The authors should also compare their ChIP seq profiles for KB cells (Fig 3A) from this paper with HeLa from ENCODE (Fig S1) to determine how comparable the two cell lines are at the chromatin level.

Reply: The IL-1-regulated enhancers flanking the *IL8* and *CXCL2* chemokine loci have been extensively compared between both, HeLa and KB cells in work from our lab published in Jurida et al., 2015. We used ChIP-seq (for KB cells) and Chip-qPCR (for HeLa and KB cells) to show that all essential features such IL-1-inducible H3K27acetylation, inducible p65 binding and TAK inhibition are shared or at least substantially overlapping between the two cell lines. Nonetheless, visual inspection of the ENCODE ChIP-seq data from the HeLa S3 line versus our KB data, shows that signal distribution is not identical (also confirming the fact that these two cells lines are disparate). Now, as regards to i4C comparison, despite the overall interaction anchors (i.e. enhancers, promoters, CTCF-bound sites) being very similar between the two lines (as one would expect given the ubiquitous nature of inflammatory responses), we would not systematically use it as a readout of cell line similarity or compatibility. In brief, there are confounding factors in 3C-based studies, and especially in 4C where strong PCR amplification is used, that preclude the fully quantitative assessment of interaction profiles, especially between cell types. Our experience says that different cell types will almost invariably have different responses to fixation, to restriction endonuclease activity, even to overall ligation efficiencies in *cis* versus in *trans*. Thus, although we were happy to see that HeLa and KB i4C interactomes converge as regards enhancer-promoter interactions, at high (individual fragment) resolution such interactomes are not expected to be identical and should not be compared on a “peak by peak” basis.

4) While HeLa cells and other cancer cell lines have been a valuable tool in several fields, their genomic composition tends to differ so much from that of primary cells that it is reasonable to question their utility for functional genomics studies. In the specific case of HeLa cells, aneuploidy, large structural variants, and even chromothripsis have been well documented. The manuscript does not provide evidence that the main CXCL cluster in 4q, in which much of the work is centered, is not affected by large structural variants in the cells they have studied. In the absence of experimental evidence, this serious limitation of cancer cell lines should at least be presented and discussed, as it affects the generalizability of the findings reported.

Reply: The reviewer raises a rather important point that should be taken into account in every study involving a cancer cell genome. To address this we had already taken precautionary steps by looking into publicly-available HeLa Hi-C data from the extended *IL8/CXCL2* locus on Chr. 4 (**Fig. EV10**). Such structural abnormalities as extensive breakage, translocations, duplications, deletions, etc. are readily detectable in Hi-C maps as aberrant signal distribution (see (Dixon, Xu et al., 2018)). As is obvious by the Hi-C data in **Fig. EV10** there are no aberrations in the loci we are investigating by i4C. In addition, we have now added additional box plots in **Fig. 3C** showing the expression values of all genes in chromosome 4. This data reveal no obvious defects in gene expression, arguing against excessive chromosomal deterioration, and so do the extensive intronic RNA FISH experiments shown in **Fig. 5** with any obvious evidence for defects of this kind compared to diploid non-transformed RPE-1 cells. Furthermore, 3D-DNA FISH using two large probes directed against the *CXC* chemokine locus and another (housekeeping) region of chr.4 does not reveal major differences in chromosomal copy numbers or translocations comparing parental HeLa, vector controls or $\Delta p65^{eIL8}$ cells (see **Fig. EV3D-E**).

5) In the intronic RNA FISH experiments, the authors report that either of the two enhancer microdeletions introduced by CRISPR led to reduced CXCL2 and IL8 transcription after IL-1 stimulation, without evidence of the hierarchical relationship between the two enhancers that has been suggested by the i4C experiments. They then make an attempt at stratifying the signal by allele-specific patterns of expression. The way in which this was done needs to be more clearly explained. As currently written, it is unclear to what extent the co-localization of signals from CXCL2 and IL8 provides a reliable and quantitative method for measuring allele-specific expression. Unless this can be shown convincingly, an alternative method that involves measurement of expressed sequence variants would be necessary to draw meaningful conclusions about differential effects of the two enhancer microdeletions.

Reply: In line with the general literature on transcriptional regulation we observe that transcription of the cytokine-responsive genes is largely stochastic and monoallelic. Assuming that HeLa cells have 3-4 copies of chromosome 4 on average per cell (see our own 3D-DNA FISH data), the intronic RNA FISH data shown in **Fig. 5B** indicate that in the majority of cells only half of all alleles are activated by IL-1 at any given time. Thus, the probability of CXCL2 and IL8 pre-mRNAs being transcribed from two different alleles and also overlapping as RNA FISH foci under the microscope in the whole of the nuclear volume is infinitely low (in less than 1 in 10^6 cells for a diploid human nucleus; modeled in (Papantonis, Kohro et al., 2012)). As a result, it is safe to assume that the vast majority of overlapping CXCL2 and IL8 signals will actually originate from activation of both genes on the same allele. If one now compares the fraction of cells showing such concomitant activation (i.e. overlapping “yellow” signal) in wild-type or control versus $\Delta p65^{eIL8}$ cells, there is a statistically-significant reduction (on top of the overall response dampening). However, if one makes this same comparison between wild-type or control versus $\Delta p65^{eCXCL2}$ cells, the difference does not appear significant. As a true negative control, cells carrying a RELA deletion show virtually no such overlapping signal (**Fig. 5C**). These results do corroborate the hierarchical relationship between the two enhancer elements studied, with the IL8 enhancer being the dominant one and also reducing CXCL2 expression from the same allele/TAD. Still, the reviewer is correct in asking this to be clarified and we elaborate on this in the relevant section of the Results.

6) The effects of enhancer knockout on genes both within and beyond the CXCL locus is striking. From an inflammation regulation perspective, it would be important to confirm that the effects of diminished gene induction in enhancer deleted cells are reflected in secreted cytokine levels. The cytokines both within (IL8, CXCL1, 2 and 3) and outwith the locus (IL6, CCL20) could be readily measured with commercially available detection kits.

Reply: This point is valid and also overlaps with comment 1 (above). We have performed ELISA experiments for secreted IL-8 and IL-6. These data nicely confirm the observations at the mRNA levels (new **Fig. 3D**). In fact, the suppression of IL-8 and IL-6 secretion by the IL8 enhancer mutant is comparable to depletion of p65 (see new **Figs 3D** and **EV6C**). Additionally, we show that the secretome (as assessed by puromycinylation of newly-synthesized polypeptides, as well as by silver staining) remains unchanged in the $\Delta p65^{eIL8}$ and $\Delta RELA$ cells confirming the specificity of these enhancers' functions (**Fig. 3E**). We now also performed cytokine arrays to reveal additionally secreted proteins, which may contribute to the autocrine/paracrine effects of IL8 enhancer mutation or p65 deletion. Despite these arrays being significantly less sensitive and only semi-quantitative compared to ELISA (see **Fig. EV6**), we were able to identify MIP-3 α (CCL20) as a third secreted factor that is downregulated in the IL8 enhancer mutant. We discuss these results in the light of the (limited) literature on IL-8 acting as an autocrine/paracrine factor in acute cytokine signaling.

7) The authors should comment on why the enhancer KO cells appear to show substantially increased i4C contacts compared to controls (Fig4 IL8 enhancer plots). There is a very brief mention that the CXCL2 enhancer deleted cells have 'rich interactomes', and no mention for the IL8 enhancer, and it is not noted for either case that the contact numbers actually go up. How would the authors explain master enhancer deletions causing increased levels of chromatin contacts within the TAD?

Reply: The reviewer is right that there is a trend for increased overall i4C signal in some of our enhancer deletion genotypes. There have been different, locus-specific views of such effects in recent (albeit sparse) literature, but we interpret this as follows: in the absence of NF- κ B binding the CXCL2 and IL8 enhancers can now be promiscuously driven to non-productive interactions of lower intensity, and this contributes to less focused and more “noisy” interactomes also indicative of increased heterogeneity in response to IL-1 stimulation in these cells. We appreciate this comment, and we have now added this explanation to the revised manuscript text.

8) The paper is severely lacking in terms of data access and availability. I don't consider it reasonable practice to state that data will be made available 'upon paper acceptance', as it prevents appropriate peer review of the manuscript data presentation and transparency.

Reply: We sincerely apologize for this shortcoming; we would never object to reviewers having direct access to raw or processed data of any sort from a manuscript under review – and this was supposed to be easily doable via download links from our servers in Giessen and Cologne. In fact, i4C profiles are small enough to also be distributed via email upon request. Nonetheless, all ATAC-seq and CHIP-seq from KB cells have been uploaded as additional datasets complementing previously published work (from Jurida et al., 2015) in GEO under the accession number GSE134436 (<https://www.ncbi.nlm.nih.gov/geo/query/acc.cgi?acc=GSE134436>, Reviewer access token: qnoluqkqbjcvtcv). All i4C data were uploaded to the SRA repository and are already publicly available under the accession number PRJNA552438 (<https://www.ncbi.nlm.nih.gov/Traces/study/?acc=PRJNA552438+&go=go>). Finally, microarray data from **Fig. 4** are also provided for convenience as an additional **Table EV1**.

Minor points:

1) Fig 1B would be more easily interpreted as a 3-way Venn diagram. This would determine how similar the no IL-1 condition is to IL-1 treated cells in presence of the TAK1 inhibitor.

Reply: we have now replaced the previous graphs with the requested 3-way Venn diagram in **Fig. 1B**.

2) Fig 2 is not so central to the paper and should be a supplementary figure. Similarly, Fig S9 is more important and would be better as a main figure.

Reply: As suggested, we have now moved **Fig. 2** to the supplement (Fig. EV1), but would also like to keep Fig. S9 (now **Fig. EV13**) as a supplementary figure to maintain focus on the mechanistically dissected *IL8/CXCL2* locus.

References cited above:

- Adey A, Burton JN, Kitzman JO, Hiatt JB, Lewis AP, Martin BK, Qiu R, Lee C, Shendure J (2013) The haplotype-resolved genome and epigenome of the aneuploid HeLa cancer cell line. *Nature* 500: 207-11
- Bird TA, Sleath PR, deRoos PC, Dower SK, Virca GD (1991) Interleukin-1 represents a new modality for the activation of extracellular signal-regulated kinases/microtubule-associated protein-2 kinases. *J Biol Chem.* 266: 22661-70
- Jurida L, Soelch J, Bartkuhn M, Handschick K, Muller H, Newel D, Weber A, Dittrich-Breiholz O, Schneider H, Bhujju S, Saul VV, Schmitz ML, Kracht M (2015) The Activation of IL-1-Induced Enhancers Depends on TAK1 Kinase Activity and NF-kappaB p65. *Cell Rep.* 10: 726-739
- Landry JJ, Pyl PT, Rausch T, Zichner T, Tekkedil MM, Stutz AM, Jauch A, Aiyar RS, Pau G, Delhomme N, Gagneur J, Korbel JO, Huber W, Steinmetz LM (2013) The genomic and transcriptomic landscape of a HeLa cell line. *G3 (Bethesda)* 3: 1213-24
- Manna SK, Ramesh GT (2005) Interleukin-8 induces nuclear transcription factor-kappaB through a TRAF6-dependent pathway. *J Biol Chem.* 280: 7010-7021
- Papantonis A, Kohro T, Baboo S, Larkin JD, Deng B, Short P, Tsutsumi S, Taylor S, Kanki Y, Kobayashi M, Li G, Poh HM, Ruan X, Aburatani H, Ruan Y, Kodama T, Wada Y, Cook PR (2012) TNF α signals through specialized factories where responsive coding and miRNA genes are transcribed. *EMBO J.* 31: 4404-14
- Saklatvala J, Kaur P, Guesdon F (1991) Phosphorylation of the small heat-shock protein is regulated by interleukin 1, tumour necrosis factor, growth factors, bradykinin and ATP. *Biochem J.* 277: 635-42

Thank you for submitting your revision to The EMBO Journal. Your study has now been re-reviewed by the two referees and their comments are provided below. As you can see, both referees appreciate the introduced changes and support publication here. It would be good to include the suggested comments made by referee #2

There are a few editorial points that need to be resolved as well.

REFeree REPORTS:

Referee #1:

In the revised version of the manuscript entitled "Hierarchical IL-1 α -responsive enhancers via acute and coordinated changes in chromatin topology" the authors have addressed all points that I have raised in the original review and have modified the manuscript accordingly. In my opinion the revised version of the manuscript is now suitable for publication at EMBO J.

Referee #2:

The revised manuscript by Weiterer et al supports my prior opinion that this is an important and insightful study of enhancer function in the human CXCL/IL8 inflammatory gene locus. The authors have improved an already impressive paper in their efforts to address my comments on the initial manuscript.

In particular they provide important supporting data comparing the HeLa and KB cells used in their study (new Figs EV2 and EV3), emphasizing the close relationship between the cell lines but also some differences that would be expected of immortalized cancer cell clones.

They also confirm the effects of enhancer deletion on protein secretion of cytokines (Fig 3C), and extend this to a broader cytokine panel in Fig EV6, and secretome analysis in Fig 3E.

They also include important discussion of their findings in the context of a recently published study of lncRNA function in the same inflammatory locus (Fanucchi et al).

My only minor point on the revised manuscript regards the new Fig 3C. The blue and green boxes to the right of this figure should either be labeled to denote the different enhancer deletions or removed from the figure. If the authors choose to keep these boxes they should consider adding legend details for the red and black conditions also included in the Fig 3C boxplots.

Overall the authors revisions have improved the manuscript considerably and I consider it suitable for publication in the EMBOJ.

Reviewers' comments:

Referee #1:

In the revised version of the manuscript entitled "Hierarchical IL-1 α -responsive enhancers via acute and coordinated changes in chromatin topology" the authors have addressed all points that I have raised in the original review and have modified the manuscript accordingly. In my opinion the revised version of the manuscript is now suitable for publication at EMBO J.

We thank this reviewer for his positive assessment and endorsement of our work.

Referee #2:

The revised manuscript by Weiterer et al supports my prior opinion that this is an important and insightful study of enhancer function in the human CXCL/IL8 inflammatory gene locus. The authors have improved an already impressive paper in their efforts to address my comments on the initial manuscript. In particular they provide important supporting data comparing the HeLa and KB cells used in their study (new Figs EV2 and EV3), emphasizing the close relationship between the cell lines but also some differences that would be expected of immortalized cancer cell clones. They also confirm the effects of enhancer deletion on protein secretion of cytokines (Fig 3C), and extend this to a broader cytokine panel in Fig EV6, and secretome analysis in Fig 3E. They also include important discussion of their findings in the context of a recently published study of lncRNA function in the same inflammatory locus (Fanucchi et al). My only minor point on the revised manuscript regards the new Fig 3C. The blue and green boxes to the right of this figure should either be labeled to denote the different enhancer deletions or removed from the figure. If the authors choose to keep these boxes they should consider adding legend details for the red and black conditions also included in the Fig 3C boxplots. Overall the authors revisions have improved the manuscript considerably and I consider it suitable for publication in the EMBOJ.

We thank this reviewer for his positive assessment and endorsement of our work. We also removed the boxes in Fig. 3C that were inadvertently left in the figure panel – we thank the reviewer for noticing this.

Altogether, we hope these changes comply with EMBO Journal requirements and can expedite processing of our manuscript.

3rd Editorial Decision

1st Oct 2019

Thanks for sending me the revised version. I have now looked at everything and all looks good. I am therefore very pleased to accept the manuscript for publication here

Corresponding Authors Names: Michael Kracht and Argyris Papatontis

Manuscript Number: EMBOJ-2019-101533